# Optogenetic rewiring of thalamocortical circuits to restore function in the stroke injured brain

Kelly A. Tennant[1], Stephanie L. Taylor[1], Emily R. White[1] & Craig E. Brown[1,2,3]

To regain sensorimotor functions after stroke, surviving neural circuits must reorganize and form new connections. Although the thalamus is critical for processing and relaying sensory information to the cortex, little is known about how stroke affects the structure and function of these connections, or whether a therapeutic approach targeting these circuits can improve recovery. Here we reveal with *in vivo* calcium imaging that stroke in somatosensory cortex dampens the excitability of surviving thalamocortical circuits. Given this deficit, we hypothesized that chronic transcranial window optogenetic stimulation of thalamocortical axons could facilitate recovery. Using two-photon imaging, we show that optogenetic stimulation promotes the formation of new and stable thalamocortical synaptic boutons, without impacting axon branch dynamics. Stimulation also enhances the recovery of somatosensory cortical circuit function and forepaw sensorimotor abilities. These results demonstrate that an optogenetic approach can rewire thalamocortical circuits and restore function in the damaged brain.

[1] Division of Medical Sciences, University of Victoria, Victoria, British Columbia, Canada V8P 5C2. [2] Department of Biology, University of Victoria, Victoria, British Columbia, Canada V8P 5C2. [3] Department of Psychiatry, University of British Columbia, Vancouver, British Columbia, Canada. Correspondence and requests for materials should be addressed to C.E.B. (email: brownc@uvic.ca).

To regain sensory and motor function after stroke, surviving neural circuits must reorganize and make new connections[1,2]. Therapies that promote better neural rewiring ultimately result in better functional outcomes[3]. Thus far, most research has focused on understanding and manipulating cortical circuits to improve recovery from ischaemic stroke[4–7]. This focus is partly born out of the fact that evoked cortical activity in the damaged hemisphere is depressed for many weeks after focal stroke[7–9], when sensorimotor deficits are most pronounced. However, sensory-evoked cortical activity is dependent on the integrity of afferent inputs from the thalamus. Despite this crucial role, it has long been overlooked as a possible substrate for mediating stroke recovery. This is somewhat surprising given that the structure and function of thalamocortical circuits can be modified well into adult life[10–15], thereby raising the possibility that these connections could be targeted and manipulated by a stroke therapy to help restore lost functions. Of the few studies that have examined thalamic circuits, it has been shown that focal ischaemia attenuates neural excitability in thalamic nuclei and leads to scattered cell death and gliosis[16–19]. The paucity of data on this subject leaves many important questions unanswered. For example, how does stroke impact sensory-driven response properties within thalamocortical axons? To what extent are thalamocortical axons capable of sprouting new branches after stroke or remodelling synaptic contacts on existing branches? Lastly, if thalamocortical connections are damaged and not functioning properly, can we target these connections with a stroke therapy to improve recovery of sensorimotor function?

The ability to manipulate the activity and wiring of specific neural circuits in vivo was not possible with previous brain stimulation technologies (that is, electrical or transcranial magnetic stimulation) and therefore has only recently been realized with the development of optogenetic approaches and viral-based gene therapy[20–22]. The use of optogenetics to treat brain pathology is still in its infancy, but it has shown promise in treating retinal degenerative disease[23], epilepsy[24], pain[25] and psychiatric disorders[26]. Not surprisingly, there is tremendous interest in the idea of using optogenetics to treat brain circuits damaged by stroke[27]. So far, only a couple of studies have attempted this, and both focused on relatively brief (3–10 days stimulation) and local modulation of channelrhodopsin (ChR2)-expressing motor circuits (that is, layer 5 motor cortex or striatal neurons) through fibre optic probes[28,29]. Although both of these important studies reported a modest improvement in gross motor behaviour and correlative changes in neurotrophin gene expression and neurovascular coupling, precisely how optogenetic stimulation impacted the wiring of damaged circuits remains unknown.

In the present study, our first goal was to understand how stroke disrupts thalamocortical connections. We show that stroke leads to a loss of axonal synaptic contacts in peri-infarct cortex and a long-lasting dampening of thalamocortical bouton excitability. Given these deficits, we hypothesized that chronic transcranial window optogenetic stimulation of thalamocortical axons could promote adaptive circuit rewiring and the restoration of sensory function. Our experiments reveal that optogenetic stimulation promotes the formation of new and stable thalamocortical synaptic boutons without impacting axonal branch dynamics. Furthermore, stimulated mice display more complete recovery of sensory cortex and paw function that persists after stimulation has ceased. Overall, our study demonstrates that chronic optogenetic stimulation can induce thalamocortical circuit rewiring and the restoration of function after brain injury.

## Results

**Thalamocortical axons are less responsive after stroke.** Since the peri-infarct region is critical for the recovery of sensorimotor functions[7,30–32], we focused our experiments on this area after focal photothrombotic stroke in the primary forelimb somatosensory cortex (FLS1). Our first goal was to determine to what extent thalamic projections to FLS1 (Fig. 1a) survive focal stroke as previous studies have suggested that these connections are disrupted[18]. We injected adeno-associated virus (AAV) into the ventral thalamus (for summary of injection sites, see Supplementary Fig. 1) and imaged in vivo enhanced green fluorescent protein (eGFP)-expressing thalamocortical axons in layers 1–3 of FLS1 cortex (Fig. 1b–c). On the basis of the location of injection sites and known patterns of connectivity, most thalamocortical projections to S1FL likely arose from the ventral posterior lateral (VPL) nucleus and the posterior medial (PoM) nucleus (Fig. 1a). One week after stroke, there was a significant reduction ($\sim 25\%$) in the density of axons in peri-infarct cortex relative to baseline (BL) ($P < 0.01$, one-way ANOVA), which remained significantly below BL levels for the remaining 4 weeks of imaging (Fig. 1c,d; $P = 0.22$, one-way ANOVA across post-stroke time points). Histological analysis of the thalamus after stroke did not reveal any obvious signs of neuronal degeneration or cell loss (Supplementary Fig. 2), but some reactive gliosis was evident.

Since most thalamic projections to peri-infarct cortex survive stroke, we then wanted to test their responsiveness to sensory stimuli. Five weeks before stroke, mice were injected with AAV to drive expression of GCaMP6s for calcium imaging and mCherry to visualize axon structure (Fig. 2a). To verify proper labelling of thalamic projections, the FLS1 and hindlimb sensory cortex were mapped with intrinsic optical signal (IOS) imaging before stroke (Fig. 2b) and 4–6 weeks afterwards (Supplementary Fig. 3). Mice without extensive axon labelling within FLS1 cortex were excluded from the study. At weekly intervals, we imaged forepaw-evoked calcium signals in thalamocortical boutons targeting layers 1–2 of the peri-infarct cortex (1,706 boutons in 4 sham stroke mice and 2,825 boutons in 6 stroke mice; Fig. 2b). Before stroke or sham procedure, 25% of GCaMP6s-expressing thalamocortical boutons responded to vibrotactile stimulation (1.5 s at 100 Hz) of the contralateral forepaw (see Fig. 2b–c). Responses to contralateral hindlimb stimulation were relatively minimal ($4.1 \pm 2.4\%$, 6 mice, 518 boutons). After focal stroke, the fraction of forelimb responsive boutons (normalized to the total number of boutons present each week) dropped at 1 week (Fig. 2c; $6.5 \pm 2.1\%$; $P < 0.05$, one-way ANOVA). Values remained significantly below BL levels for up to 4 weeks after stroke (Fig. 2c; 2 weeks $= 13.1 \pm 4.5\%$, 3 weeks $= 13.2 \pm 1.9\%$, 4 weeks $= 11.1 \pm 1.1\%$). Stroke did not significantly alter the latency of forelimb-evoked responses (Supplementary Fig. 4a; $P = 0.45$, one-way ANOVA). The peak amplitude of responses (normalized to BL values) did drop 1–2 weeks after stroke and then increased by 3–4 weeks of recovery (Supplementary Fig. 4b). Mice subjected to sham stroke did not show any significant changes in the fraction of forelimb responsive boutons over time (Fig. 2c; $P = 0.96$ based on one-way ANOVA), or changes in the latency or peak amplitude of responses (Supplementary Fig. 4; $P > 0.05$ for both groups based on one-way ANOVA). Of note, normalized axonal GCaMP6s fluorescence ($F_0$) did not change significantly over the 5-week imaging period for both groups (Fig. 2d; $P > 0.05$ based on one-way ANOVA).

Although we were generally unable to longitudinally track calcium signals from the exact same boutons, there were a few exceptions. As shown in Fig. 2e, the formation of a new bouton was accompanied by the presence of a sensory-evoked calcium transient. These examples are consistent with previous

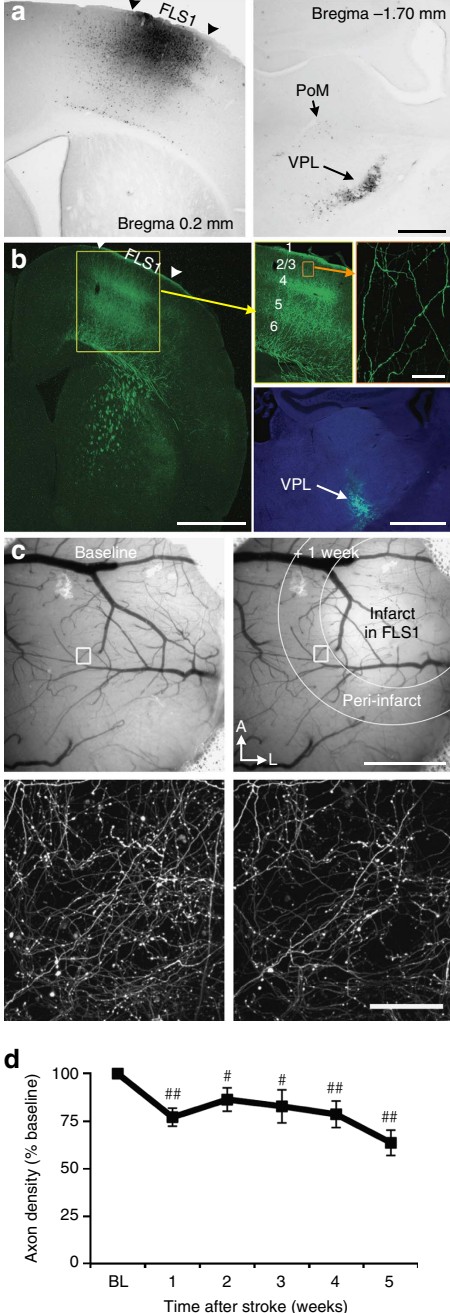

**Figure 1 | Survival of peri-infarct thalamocortical projections after focal stroke.** (**a**) Left, bright-field image shows cholera toxin B injection site in primary forelimb somatosensory cortex (FLS1). Right, retrograde labelling of neurons is primarily concentrated in the VPL nucleus, with sparse labelling in PoM nucleus of the thalamus. (**b**) Distribution of anterogradely labelled eGFP-expressing thalamocortical axons in the FLS1 cortex after AAV injection in the ventral thalamus (bottom right). (**c**) Bright-field images of the cortical surface before and 1 week after focal stroke in FLS1 cortex. Below each image is a maximal intensity z-projection two-photon image of thalamocortical axons in cortical layer 1 of FLS1/peri-infarct cortex. (**d**) Graph shows the density of axons after stroke relative to BL levels ($n = 15$ mice; [#]$P < 0.05$; [##]$P < 0.01$ relative to BL based on one-sample $t$-tests). Data are means ± s.e.m. Scale bars, (**a**) 0.5 mm; (**b**) 1 mm (top left), 50 μm (top right), 1 mm (bottom right); (**c**), 1 mm (top right), 50 μm (bottom right).

studies[33,34] showing that new axonal boutons are presumably functional.

**Stimulation activates cortical circuits but not blood flow.** The dampening of thalamocortical axon excitability fits with previous studies showing impaired superficial cortical layer responses to sensory stimuli in the stroke-affected hemisphere[35,36]. In light of these findings, we hypothesized that boosting thalamocortical excitability with an optogenetic approach could enhance the return of sensory function. However, before testing this idea, we conducted several experiments to validate our approach. First, we verified that AAVs could drive ChR2 and mCherry expression in thalamocortical projections (Fig. 3a). As expected, axonal projections terminated in cortical layers 1, 4 and 5 (Fig. 3b). To deliver chronic optogenetic stimulation, we used transcranial window light stimulation from a removable magnetic head mounted light-emitting diode (LED) (Fig. 3c). This approach allowed us to stimulate freely moving mice in their home cage (inset in Fig. 3c). To determine which light power density to employ, we recorded forepaw and light-evoked cortical field potentials *in vivo*. We determined that a 475 nm blue 5 ms light pulse at $10 \, mW \, mm^{-2}$ reliably activated cortical neurons in a CNQX/AP5 sensitive manner that also closely resembled the cortical response amplitude to a 5 ms deflection of the forepaw (Fig. 3d–e). This power density is similar to previous studies that employed a head mounted LED to optogenetically activate cortical neurons *in vivo*[37,38]. For optical stimulation, we delivered 5 ms light pulses at 5 Hz (1 s ON/4 s OFF; Fig. 3f) for 60 min per day (3,600 pulses per day). Five hertz was selected because it is within the normal frequency range of thalamic neuron firing and mimics the natural frequency of mouse limb strides[39,40]. Similar to mechanical 5 Hz stimulation of the forepaw, the peak amplitude of optically evoked cortical potentials decreased with successive pulses (Fig. 3f and Supplementary Fig. 5a). However, the average amplitude of optically evoked responses after 60 min of stimulation was not significantly different from BL levels (Supplementary Fig. 5b). Further, 60 min of stimulation did not generate any signs of axonal damage (Supplementary Fig. 5c).

To determine if optogenetic stimulation influenced cortical blood flow, we generated laser speckle contrast maps before, during and after our 60 min stimulation protocol (Fig. 3g). Our data show that intermittent 5 Hz optogenetic or control stimulation (blue light with no ChR2 expression) of thalamic axons had little impact on cerebral blood flow (Fig. 3g,h; $P > 0.05$ for both groups based on ANOVA). To prove our methods were sufficiently sensitive, we conducted two positive control experiments. First, we replicated previous work by Steinberg's group[28] and found that continuous 10 Hz stimulation of ChR2-expressing thalamic axons for 60 s led to a 17 ± 3% increase in blood flow (Fig. 3g,h). Second, we challenged mice with 6% $CO_2$ exposure and found a significant increase in cerebral blood flow (Fig. 3g,h). Together, these experiments show that our low intensity regime of optical stimulation activated thalamocortical circuits but did not induce detectable changes in regional blood flow.

**Stimulation promotes rewiring of thalamocortical boutons.** To determine if chronic stimulation can induce thalamocortical circuit rewiring after stroke, thalamocortical axons were imaged *in vivo* at weekly intervals (Fig. 4a). We should note that since the AAV only infected neuronal cell bodies in the thalamus, we did not encounter fluorescently labelled cortical dendrites when imaging. This mitigated any confusion in distinguishing beaded presynaptic versus ischaemic post-synaptic structures. Optogenetic stimulation commenced 3 days after stroke or sham procedures to

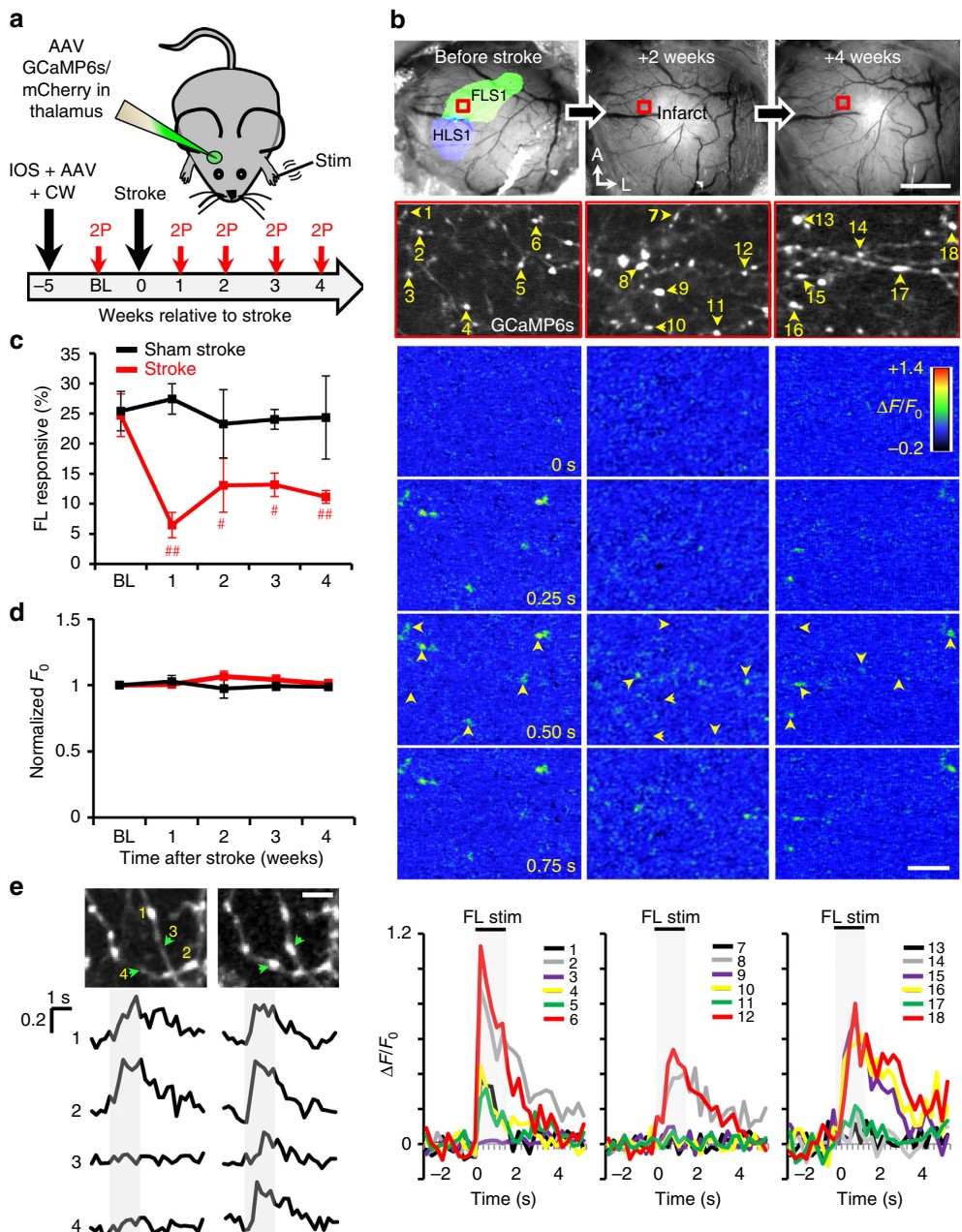

**Figure 2 | Thalamocortical axons are less responsive to forelimb stimulation after stroke.** (**a**) Graphical summary of GCaMP6s imaging experiments. (**b**) Top, bright-field images of the cortical surface before and after stroke (top row). IOS imaging was used to delineate the functional representation of the FLS1 and primary hindlimb somatosensory cortex. Below; images showing GCaMP6s-expressing thalamocortical boutons (numbered 1–18) within boxed regions. Colour montages below show forelimb-evoked changes in GCaMP6s fluorescence ($\Delta F/F_0$) for axonal boutons imaged before stroke and 2–4 weeks later. Plots show calcium responses to forelimb stimulation ($\Delta F/F_0$) within individual boutons at each time point. (**c**) Graph shows the percentage of forelimb responsive thalamocortical boutons plotted as a function of time after stroke (2,825 boutons were analysed in 6 stroke mice) or sham (1,706 boutons analysed in 4 mice) procedure. (**d**) Normalized GCaMP6s fluorescence ($F_0$) in axonal boutons over time. (**e**) Top, images show mCherry-labelled thalamocortical boutons (numbered 1–4) imaged 1 week apart. Bottom, shows forelimb evoked responses ($\Delta F/F_0$) within each numbered bouton. Note that newly formed boutons (#3, 4) show detectable forelimb-evoked calcium responses. Data are means ± s.e.m. #$P<0.05$; ##$P<0.01$ based on $t$-tests comparing BL to each week of recovery. Scale bars, (**b**) 1mm (top right), 10 µm (bottom right); (**e**) 5 µm.

avoid overstimulating the cortex in the acute phase, which could exacerbate ischaemic damage[41]. Given the persistent decrease in thalamocortical axon excitability after stroke (Fig. 2c), stimulation continued from 3 days to 6 weeks post stroke, with five consecutive sessions per week. Stroke-affected mice that received control stimulation for 6 weeks consisted of mice that received light stimulation in the absence of ChR2 ($n=4$ mice) or expressed ChR2

but did not receive light stimulation ($n=7$ mice). There were no significant differences between the two types of control stimulation for all dependent measures tested in this study (see Supplementary Fig. 6; $P>0.05$ for each metric based on two-way ANOVA), except terminaux bouton (TB) dynamics ($P<0.01$ based on ANOVA). To simplify data presentation, both types of stimulation controls were combined and presented as one group (see purple lines representing

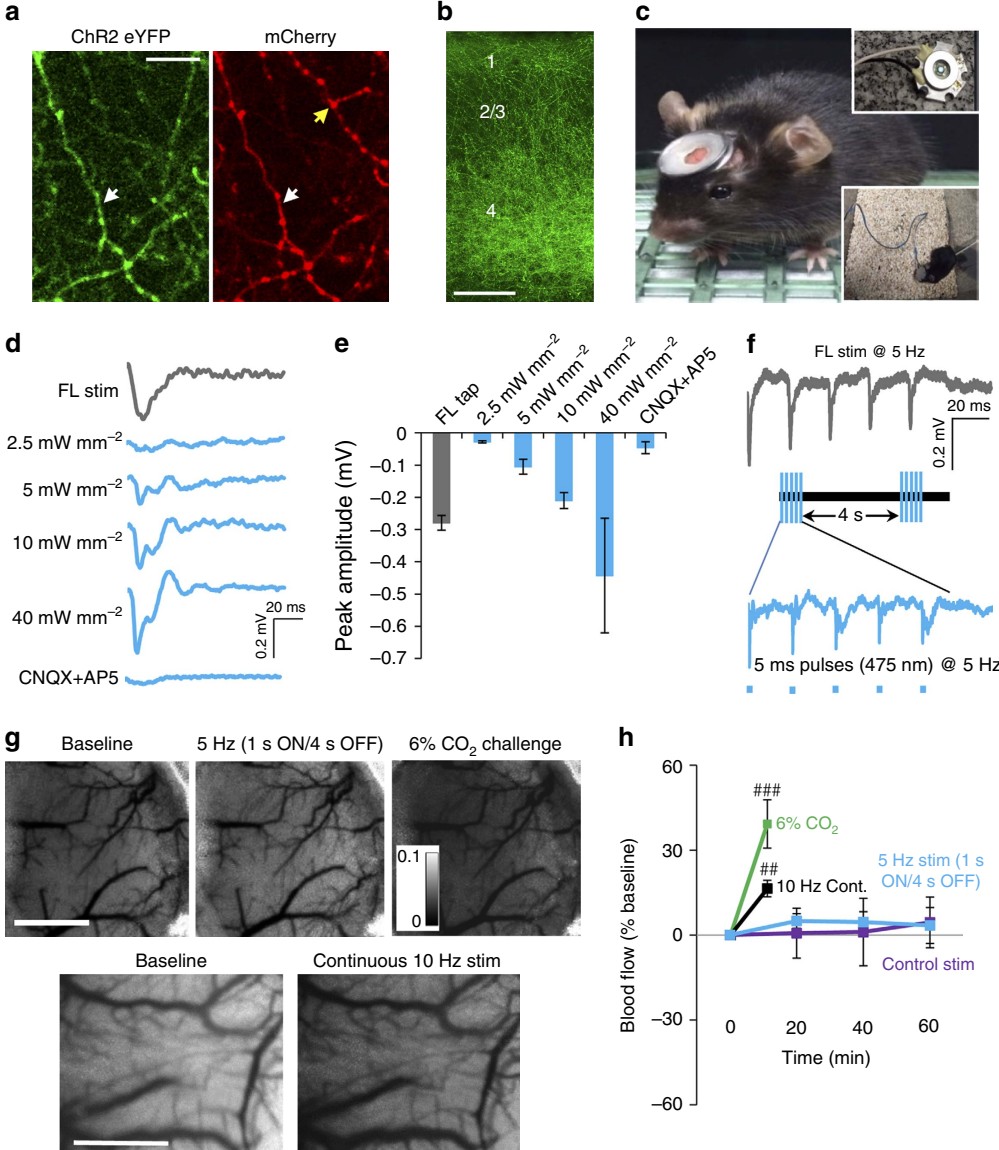

**Figure 3 | Optogenetic stimulation activates cortical circuits but not cerebral blood flow.** (**a**) *In vivo* two-photon images showing co-expression of ChR2-eYFP with mCherry in many (white arrow), but not all (yellow arrow) thalamocortical axons. (**b**) Confocal image showing laminar distribution of ChR2-eYFP thalamic axons in somatosensory cortex. (**c**) For longitudinal imaging in ChR2-stimulated mice, a cranial window was installed that could reversibly attach to a magnetic head-mounted 475 nm blue LED (max output power ∼10 mW mm$^{-2}$). (**d**) Field potentials recorded in layer 2 of somatosensory cortex (averaged over 20–30 trials) in response to a single 5 ms forepaw deflection or blue light pulse. (**e**) Graph shows peak amplitude of cortical field potentials in response to a single forepaw deflection or blue light pulse over a range of power intensities ($n = 3$ mice). (**f**) Representative field potential recordings showing cortical responses to 5 Hz trains of 5 ms forelimb or light stimulation. (**g**) Top row, laser speckle contrast images shown at BL and 60 min after intermittent 5 Hz optogenetic stimulation (1 s ON/4 s OFF for 60 min). The same mouse was exposed to 6% $CO_2$ for 90 s as a positive control (darkening of cortex with pixel values closer to 0 reflects increased blood flow). Bottom row, for an additional positive control, laser speckle images are shown at BL and immediately following continuous 10 Hz stimulation for 60 s. (**h**) Graph plots changes in cerebral blood flow over time in mice that received intermittent 5 Hz optogenetic (eight trials in four mice) or control stimulation (six trials in three mice, blue light stimulation but no ChR2 expression), continuous 10 Hz stimulation (eight trials in four mice) or 6% $CO_2$ exposure (four trials in two mice). $^{##}P < 0.01$; $^{###}P < 0.001$ based on paired *t*-tests. Data are means ± s.e.m. Scale bars, (**a**) 10 µm; (**b**), 100 µm, (**g**) 1 mm.

'Stroke + Control stim' in Figs 4–7). Our stroke control groups consisted of mice that received sham stroke surgery in tandem with 6 weeks of optogenetic stimulation or control stimulation. Although each of these sham stroke control groups is shown separately in Figs 5–7 (see solid black and dashed grey lines), there were no significant differences between these sham control groups on any dependent measure tested in this study ($P > 0.05$ for main effect of group for each metric based on two-way ANOVA).

We first examined whether optogenetic stimulation could influence the growth and retraction of thalamocortical axon branches (defined as any branch $> 5$ µm in length at first imaging session). Although stroke led to a significant reduction in axonal branch length that was evident in the first week of recovery (Fig. 4b), there was no effect of stimulation on branch growth or retraction (Fig. 4b; $P = 0.12$ for main effect of group based on ANOVA). Furthermore, while most axons were stable, we noted a few examples where a branch extended or was eliminated (Fig. 4c), suggesting that large-scale remodelling can occur as previously reported[42,43].

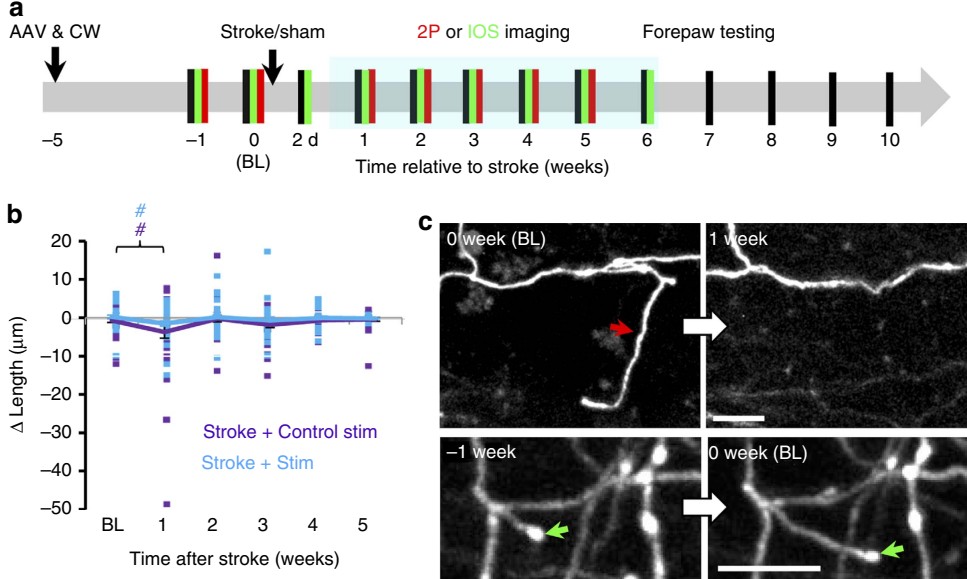

**Figure 4 | Restructuring of axonal branches after stroke.** (**a**) Timeline for experiments. Behavioural tests of forelimb function (black lines) were regularly conducted (usually at weekly intervals) in sham or stroke-affected mice that underwent two-photon imaging of thalamocortical structure (red lines) or intrinsic signal optical imaging (green lines). Behavioural tests were always conducted the day before imaging. (**b**) Graph plots the growth and retraction of thalamocortical axonal branches in stroke-affected mice with or without optogenetic stimulation (Stroke + Control stimulation: $n = 40$ branches; Stroke + stimulation: $n = 38$ branches). The average change in branch length is shown with solid lines. (**c**) In vivo time lapse images (maximal intensity z-projection) showing examples of axonal branch elimination (upper) and growth (lower). #$P < 0.05$ based on t-tests comparing post-stroke time points in each group designated by colour to respective BL values. Scale bar, (**c**) 10 μm.

Since stimulation did not induce widespread growth of axonal branches, we next focused on synaptic boutons. Synapses along the axon appear as varicosities, which can be categorized as either TB or en passant boutons (EPB)[33,44]. TBs exhibit a bulbous head connected to the parent axon by a short neck (Supplementary Fig. 7a; operationally defined as boutons that project 1–5 μm away from the parent shaft). Conversely, EPBs display the hallmark 'beads on a string' appearance (Fig. 5a), occurring as bright fluorescent swellings that exceeded two times the median fluorescence of the shaft[34]. EPBs comprised the majority of bouton type in this study (84% EPB versus 14% TB). Further, the proportion of TB relative to EPB was not affected by optogenetic stimulation or stroke (Supplementary Fig. 7b; $P > 0.05$ for main effects of stimulation and time based on two-way ANOVA). For the TB population, we noted that a fraction of these boutons retracted or formed from week to week (Supplementary Fig. 7a). For mice that received a sham stroke procedure with optogenetic stimulation ($n = 6$ mice) or without ($n = 5$ mice), TB turnover ratio (TOR) did not change significantly over time in either group (Supplementary Fig. 7c; $P > 0.05$ for main effect of time based on ANOVA for both groups). It should be noted that TB turnover ratios were quite variable owing to the fact that they were more difficult to sample than the abundant EPB. Following stroke, TB TOR was elevated relative to BL values (Supplementary Fig. 7c). While TB TOR returned to normal levels by 3 weeks post stroke in the control stimulation group, stimulated mice retained higher TOR for up to 5 weeks (Supplementary Fig. 7c; $P = 0.03$ for group × time interaction based on two-way ANOVA). This extended period of enhanced TB turnover in stimulated mice was largely due to increased rates of TB elimination and formation (Supplementary Fig. 7d; $P < 0.05$ based on two-way ANOVA for both TB elimination and formation). Examination of TB density levels indicated a reduction in the first week after stroke, with a slight trend towards an increase over time (Supplementary Fig. 7e). Lastly, our analysis of changes in the length of all TBs (including stable ones) indicated no effect of

stroke or stimulation (Supplementary Fig. 7f). Thus, optogenetic stimulation alters TB dynamics in stroke-affected mice primarily at later stages of recovery (3–5 weeks).

Next, we asked whether optogenetic therapy influenced the dynamics of the more abundant population of EPBs (Fig. 5a). In sham stroke controls that received optogenetic stimulation ($n = 6$ mice) or not ($n = 5$ mice), there were no significant changes in EPB TOR over time (Fig. 5b; $P = 0.14$ for main effect of time based on two-way ANOVA). However, rates of EPB formation and elimination did change over time in sham stroke mice ($P < 0.05$ main effect of time based on two-way ANOVA for both EPB elimination and formation). It should be noted that no changes occurred in the first week after sham stroke, suggesting that the sham procedure itself was not responsible for these changes (see black and grey dashed lines in Fig. 5c). For stroke-affected mice, EPB TOR was significantly increased in both stimulated and control stimulation groups (see teal and purple lines in Fig. 5b; $P < 0.05$ main effect of time based on two-way ANOVA). Optogenetic stimulation had little impact on EPB TOR in the first week after stroke but significantly enhanced TOR at 2–3 weeks recovery relative to the control stimulation group (Fig. 5b; $P < 0.01$ group × time interaction based on two-way ANOVA). Notably, increased EPB turnover in stimulated mice with stroke was driven by the dramatic increase in EPB formation at 2–3 weeks recovery (Fig. 5c; $P < 0.01$ group × time interaction based on two-way ANOVA). The density of EPBs reflected this proliferation of boutons as levels returned to, or exceeded pre-stroke levels in stimulated mice, whereas EPB density in control stimulated mice remained consistently below pre-stroke levels (Fig. 5d). Increased EPB formation at 2–3 weeks recovery was also significantly correlated with distance from the infarct border ($r = -0.35$, $P = 0.01$), suggesting that EPB plasticity is highest in regions proximal to the infarct. It should be noted that group differences in EPB turnover cannot be explained by differences in the imaging location relative to the infarct (Stroke + control stim = $153 \pm 84$ μm versus Stroke + stimulation = $141 \pm 94$ μm; $P = 0.47$ based on t-test).

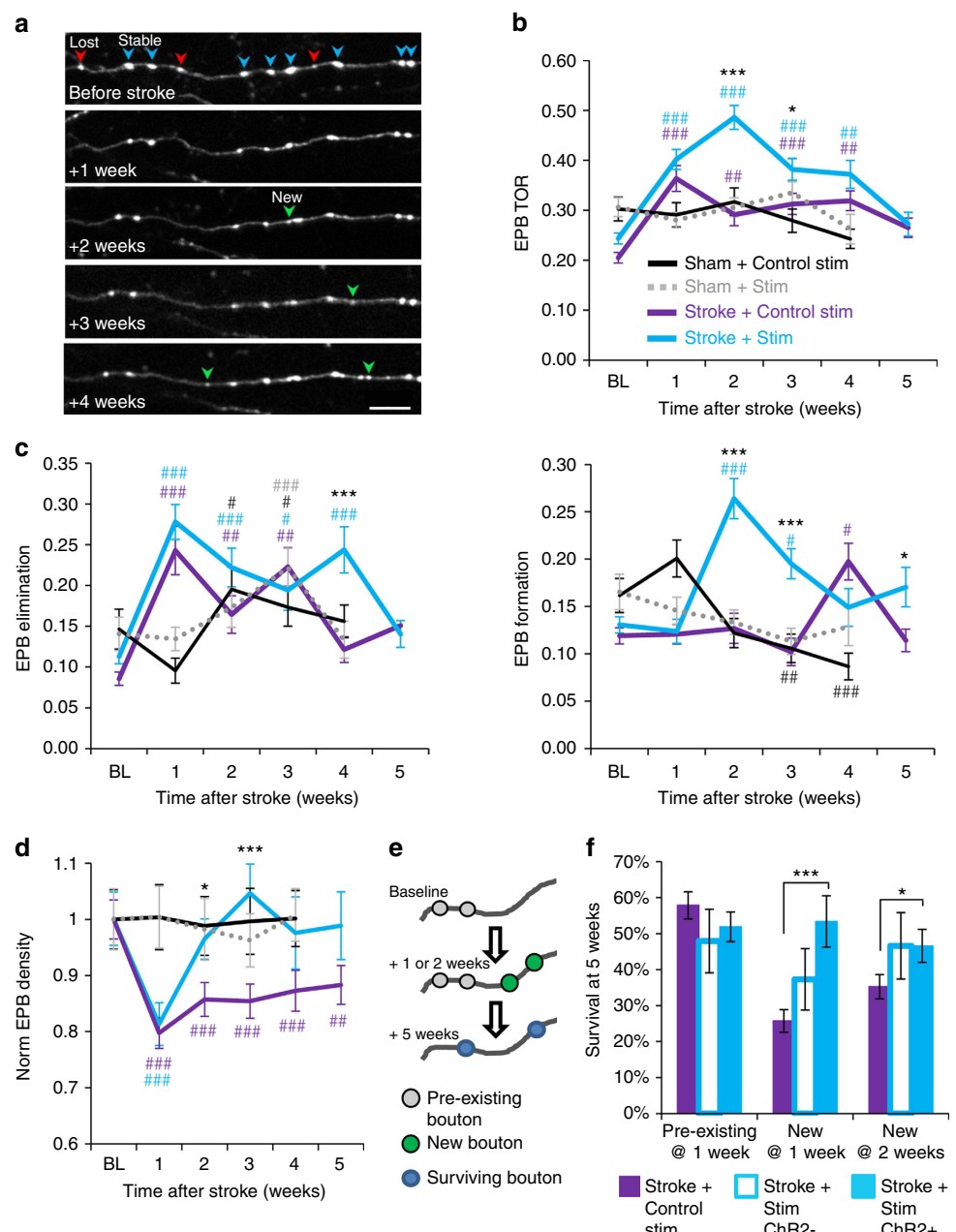

**Figure 5 | Optogenetic stimulation promotes the formation of new and stable thalamocortical boutons.** (**a**) Time lapse *in vivo* images showing the formation and elimination of thalamocortical EPBs over several weeks following stroke. Blue arrowheads = stable EPB, red arrowhead = lost EPB, green arrowhead = new EPB. (**b**) EPB TOR in mice with sham stroke (with control stimulation: n = 5 mice, 125 axons, 1,001 EPBs at BL; or ChR2 stimulation: n = 6 mice, 110 axons, 1,243 EPBs at BL), stroke with optogenetic stimulation (n = 9 mice, 101 axons, 2,423 EPBs at BL) or those with control stimulation (n = 6 mice, 90 axons, 1,477 EPBs at BL). (**c**) Graph plotting the fraction of EPB eliminated or formed each week. (**d**) Graph plotting the density of thalamocortical EPBs normalized to BL values. (**e**) Cartoon summary of our analysis of EPB survival rates (blue boutons that were present at 1, 2 weeks and 5 weeks) based on whether an EPB was preexisting (grey) or newly formed (green) at 1 or 2 weeks post stroke. (**f**) Histogram showing the survival (at 5 weeks post stroke) of preexisting versus newly formed EPB in axons from mice that received control stimulation (purple), or ChR2 negative (open teal bars) or positive axons (filled teal bars) from mice that received optogenetic stimulation. Data are means ± s.e.m. #$P < 0.05$, ##$P < 0.01$, ###$P < 0.001$ based on $t$-tests comparing post-stroke time points in each group designated by colour to respective BL values. *$P < 0.05$, **$P < 0.01$, ***$P < 0.001$ based on $t$-tests comparing stroke + stimulation versus stroke + control stimulation. Scale bar, (**a**) 10 μm.

Since the formation of new and stable synapses is a structural correlate of learning/memory[45,46] and ocular dominance plasticity[47], we next asked whether optogenetic stimulation differentially affected the survival of 'preexisting' EPBs (that is, those that existed before stroke) versus EPBs that formed after stroke (Fig. 5e). Furthermore, since some mCherry-labelled axons in our stimulated group did not express ChR2 (see Fig. 3a), we

parsed axons that were ChR2 positive or negative to determine whether stimulation effects were restricted to ChR2-expressing axons. Our analysis revealed that stimulation did not affect the survival of preexisting EPBs (Fig. 5f), but significantly enhanced the survival of new ChR2-expressing EPBs formed at 1 and 2 weeks recovery (Fig. 5f). Of note, there was also a trend towards enhanced survival of ChR2-negative EPBs in the stimulated group

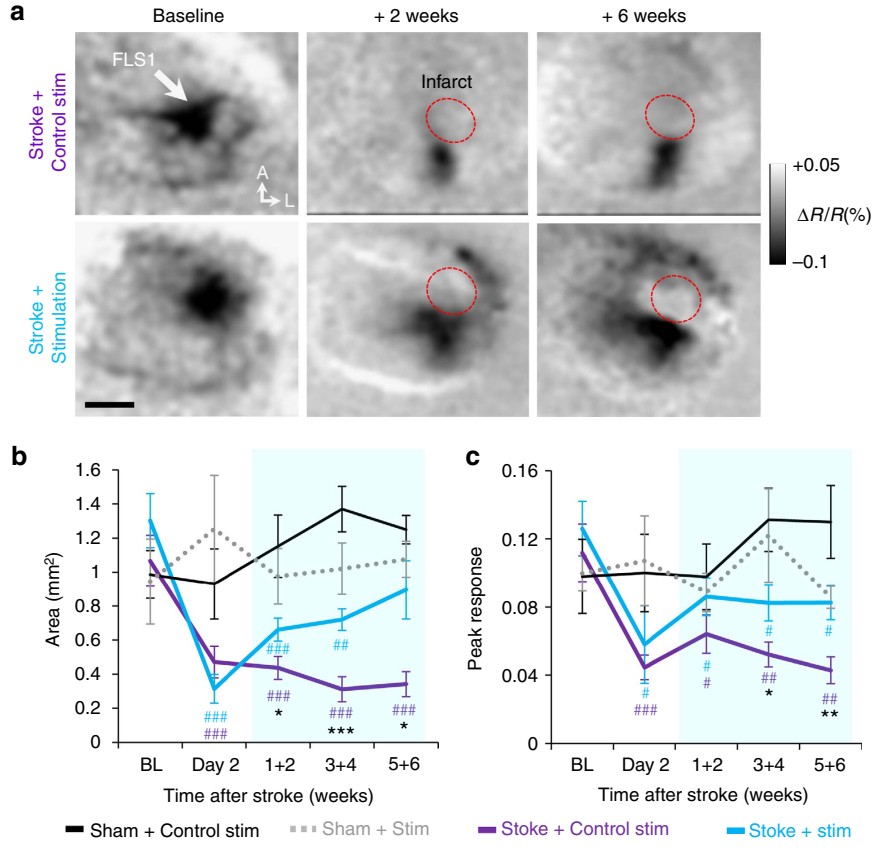

**Figure 6 | Optogenetic stimulation enhances recovery of sensory cortical circuits.** (**a**) Time lapse IOS images showing the primary somatosensory cortical response (darkened area) to stimulation of the contralateral forepaw (100 Hz for 1 s). Red circle denotes infarct area. Grey levels in each image reflect percent changes in light reflectance averaged over 0.5–1.5 s after forelimb stimulation relative to BL. (**b**) Graph plotting the area of cortical territory activated by forelimb stimulation in sham stroke controls (Control stimulation: $n = 5$ mice; ChR2 stimulation: $n = 7$ mice) or mice with stroke that received control or optogenetic stimulated mice (Control stimulation: $n = 11$ mice; ChR2 stimulation: $n = 10$ mice). (**c**) Graph plotting peak cortical responses (absolute value of $\%\Delta R/R$) to forelimb stimulation. Data are means ± s.e.m. #$P < 0.05$, ##$P < 0.01$, ###$P < 0.001$ based on $t$-tests comparing post-stroke time points in each group designated by colour to respective BL values. *$P < 0.05$, **$P < 0.01$, ***$P < 0.001$ based on $t$-tests comparing stroke + stimulation versus stroke + control stimulation. Scale bar, (**a**) 1 mm.

(new at 1 week, $P = 0.09$; new at 2 weeks, $P = 0.10$ based on $t$-tests). Collectively, these results show that optogenetic stimulation enhances the formation and stabilization of new EPBs after stroke.

**Stimulation enhances the return of sensorimotor function.** Knowing that optogenetic stimulation enhanced synaptic bouton formation and survival after stroke, we next examined what impact stimulation had on functional recovery. We used longitudinal IOS imaging to track stroke-related changes in the forepaw sensory representation (Fig. 6a, see timeline in Fig. 4a). We employed IOS imaging rather than two-photon imaging of GCaMP6s signals in axonal boutons because cortical responses can be imaged with a dim red light (635 nm; 0.5 mW mm$^{-2}$), which would not activate ChR2-expressing axons (which absorb light in the 940 nm range used for GCAMP6s imaging, but poorly at 630 nm) and thus avoid confounding our experiment[48]. For mice that underwent sham stroke procedure combined with optogenetic stimulation or without ($n = 7$ and 5 mice, respectively), the area and peak response amplitudes of cortex activated by contralateral forepaw stimulation did not change significantly over time (Fig. 6b,c; $P > 0.05$ main effect of time based on ANOVA for both map area and peak response). Two days after stroke but before the start of optogenetic therapy, mice from both stroke groups had lost a significant portion of the

forelimb map (Fig. 6a,b), which was also less responsive (Fig. 6c). For stroke-affected mice that received control stimulation, the area and peak amplitude of cortical territory activated by forepaw stimulation did not change considerably post stroke (Fig. 6a–c). However, mice that received chronic optogenetic stimulation displayed greater recovery of forelimb-evoked response amplitude (Fig. 6c; $P < 0.01$, main effect of group based on two-way ANOVA) and area (Fig. 6b; $P < 0.01$, main effect of group based on two-way ANOVA), relative to stroke-affected mice that received control stimulation.

To determine if sensorimotor function of the forepaw had also improved with optogenetic therapy, we employed two sensitive behavioural assays: the horizontal ladder walking and adhesive tape removal tests. Although there were no acute effects of sham stroke procedure on ladder performance (at 2 and 7 days post stroke), the sham stroke group that received control stimulation did show changes in forepaw placements over time (Fig. 7a; $P < 0.01$ main effect of time based on ANOVA for both correct and incorrect paw placements). Mice that received focal stroke showed a deficit 2 days post stroke, reflected by a significant drop in 'correct' forepaw placements and an increase in 'partial' placements (Fig. 7a). We should note that almost all steps taken fell into these two categories. Complete slips were only evident in 2% of steps in the first week of recovery, and did not differ between groups. However, by 2 weeks recovery and thereafter, stimulated mice showed significantly more correct placements

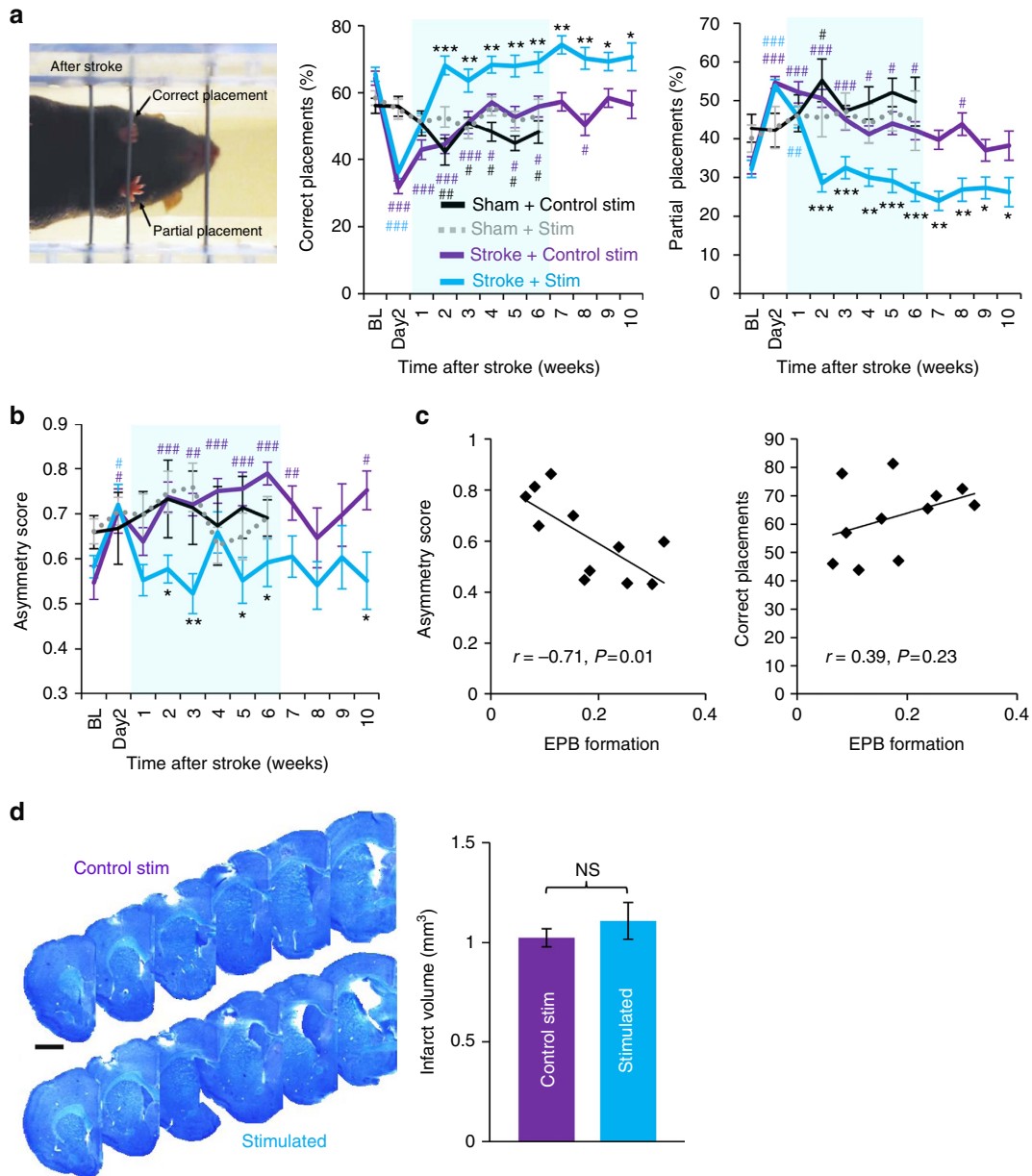

**Figure 7 | Recovery of forepaw function is enhanced by optogenetic stimulation.** (**a**) Left, image showing a mouse making a partial placement with the stroke-affected paw and a correct forepaw placement with the good limb. Centre and right, graphs showing the percentage of total placements (of the stroke affected forepaw) that were classified as either 'correct' or 'partial' in sham stroke controls (Control stimulation: $n = 5$ mice; ChR2 stimulation: $n = 6$ mice) and stroke affected mice that received control or optogenetic stimulation ($n = 8$ mice for each stroke group). (**b**) Graph shows asymmetry in tape removal latency (scores $> 0.5$ indicate more time spent removing tape from stroke affected paw) in each group (**c**) Scatter plots showing the correlation between EPB formation rates at 2 or 3 weeks recovery with their respective scores on tape and ladder tests. (**d**) Representative montages of cresyl violet stained coronal sections showing the anterior to posterior extent of focal photothrombotic stroke. Right, analysis of infarct volume in each group ($n = 6$ mice per group, $P = 0.21$ based on $t$-test). Data are means ± s.e.m. #$P < 0.05$, ##$P < 0.01$, ###$P < 0.001$ based on $t$-tests comparing post-stroke time points in each group designated by colour to respective BL values. *$P < 0.05$, **$P < 0.01$, ***$P < 0.001$ based on $t$-tests comparing stroke + stimulation versus stroke + control stimulation. Scale bar, (**d**) 1 mm.

and fewer partial placements than stroke-affected mice given control stimulation (Fig. 7a; $P < 0.001$ group × time interaction based on two-way ANOVA for both correct and partial placements). The tape removal test assesses forepaw use asymmetry[49] by determining whether the time to remove a piece of tape from each paw is equivalent (ratio of 0.5) or if there is a bias towards one paw or the other (ratio $> 0.5$ indicates more time to remove tape from the stroke-affected paw). Both sham stroke control groups did not show any significant changes over time (Fig. 7b; $P > 0.05$ for main effect of time based on ANOVA).

All stroke-affected mice showed a deficit 2 days following stroke (Fig. 7b), but only stimulated mice showed a normalization of forelimb preference/use over time relative to control-stimulated mice (Fig. 7b; $P < 0.05$ group × time interaction based on two-way ANOVA). Notably, behavioural improvements persisted after therapy ceased ($> 6$ weeks). Furthermore, there was a significant correlation between EPB formation rates at 2–3 weeks recovery with performance on the tape test and a trend towards better performance on the ladder test (Fig. 7c), suggesting that structural growth is associated with functional improvements.

Lastly, we measured infarct volume (Fig. 7d) to determine if stimulation altered ischaemic damage. Consistent with the idea that damage is largely resolved by 3 days post stroke[50], we found no differences in infarct volume between the two groups (Fig. 7d, $n = 6$ mice per group; $P = 0.21$ based on $t$-test). This result, in tandem with our data showing equivalent behavioural deficits and loss of forepaw sensory maps 2 days after stroke (before optogenetic treatment started), demonstrate that the benefits of optogenetic stimulation are not likely mediated by an early, protective effect from ischaemic damage.

## Discussion

Most studies examining the neural mechanisms underlying recovery of sensorimotor function after stroke have focused on changes to cortical neuron structure or function. For example, it is well known that stroke leads to deficits in superficial (layers 1–3) cortical responses to sensory stimuli[7,35], which are associated with changes in intracortical inhibition[8] and loss of excitatory synaptic connections[6,51]. However, whether a loss of cortical sensory responsiveness could be traced to a disruption in circuits upstream, such as those from the thalamus, is poorly understood. To help resolve this ambiguity, we first examined thalamocortical axon excitability after stroke. Our GCaMP6s imaging experiments revealed that normally, 25% of thalamocortical boutons respond to brief (1.5 s) vibrotactile stimulation of the contralateral forepaw. The proportion of responsive boutons dropped significantly 1 week after stroke and recovered slightly from 2–4 weeks. While there are no other studies examining thalamocortical axon responsiveness in somatosensory cortex (using GCaMP6) to draw direct comparisons with, our estimate of bouton responsiveness was lower than recent studies examining thalamocortical responses in layer 1 of visual cortex[52–54]. While resolving this discrepancy is beyond the scope of this study, differences in the proportion of responsive axons may reflect inherent functional differences between the sensory modalities, the location of axonal imaging (imaging near the border of a sensory representation rather than centre) or the properties of the sensory stimulus (for example, vibrotactile versus light stimuli, 1.5 versus 10 s of stimulation). Another issue that deserves discussion is the possibility that a drop in responsiveness after stroke could arise from poor imaging conditions or systematic changes to resting calcium levels. While this is a potential concern particularly in the first few hours to days after stroke when oedema is strongest[55], our finding that resting GCaMP6s fluorescence ($F_0$) in thalamocortical boutons was not affected by stroke suggests otherwise.

Our *in vivo* time lapse imaging experiments show that under control conditions, weekly turnover rates of thalamic EPBs range from 20 to 30%, with relatively balanced rates of bouton elimination and formation. Although there are no other reports on thalamocortical axon TOR in forelimb cortex, our estimates are generally in line with other *in vivo* imaging studies quantifying axonal bouton TOR from neurons in cortical layers 2/3, 5 and 6, and posterior nucleus of the thalamus[33,34]. After stroke, thalamocortical bouton TOR invariably spiked due to a loss of boutons in the peri-infarct area, similar to that described for cortical dendritic spines after stroke[5,7]. Optogenetic stimulation initiated 3 days after stroke had little effect on the initial loss of boutons, but enhanced the production of new and stable EPBs at 2–3 weeks recovery, which was contemporaneous with improved performance on behavioural tests. On the other hand, optogenetic stimulation did not influence TB formation until slightly later stages of recovery (3–4 weeks). Since the function of TBs remains a mystery, further experiments will be needed to clarify any potential role they play in stroke recovery.

Another interesting observation was that optogenetic stimulation improved IOS-based cortical responses to forelimb stimulation (both in amplitude and area) without greatly impacting axonal branch growth. This is not necessarily surprising since thalamic axons, unlike some intracortical axons, tend to exhibit less growth in the mature or damaged brain[42,56]. However, the absence of large-scale axonal growth does imply that functional changes in circuits downstream of thalamic inputs, such as intracortical circuits that propagate sensory-driven activity horizontally, are likely involved. One caveat that our study does not address is whether chronic stimulation influenced or hastened the recovery of thalamocortical axon excitability. Unfortunately since two-photon imaging of GCaMP6s also activates ChR2 in neurons[48], future studies employing red shifted opsins or genetically encoded calcium indicators would be necessary to resolve this question.

To our knowledge, this is the first study to describe the impact of optogenetic stimulation on axonal bouton dynamics *in vivo*. There are a few papers showing that dendritic spine stability can be enhanced or reduced by low frequency (3–10 Hz) optogenetic stimulation of layer 5 pyramidal neurons[38] or inhibitory somatostatin interneurons[57], respectively. Our data showing that chronic stimulation favoured the production of new synaptic boutons and their survival over existing ones is in agreement with studies imaging dendritic spine/filopodial dynamics during visual or motor learning[45,46,58]. Importantly, enhanced synaptic bouton formation correlated with better recovery of sensorimotor paw function, which mirrors previous studies where motor learning correlated with increased cortical dendritic spine production[45,46].

Our data indicate that chronic stimulation of thalamocortical sensory circuits leads to a rapid, robust and persistent enhancement of forepaw function after stroke. However, it should be noted that a few other groups have tested whether optogenetic stimulation can improve stroke recovery[28,29]. Although these other studies found a beneficial effect of stimulation on recovery, there are important differences to consider. First, our study focused on stimulating thalamocortical sensory circuits rather than those concerned with motor output, such as layer 5 motor cortex neurons or inhibitory striatal neurons. Second, we employed up to 6 weeks of transcranial stimulation as opposed to previous studies, which used 4–7 days of local stimulation delivered through a surgically implanted optrode. Therefore, our study provides new evidence that a transcranial method of stimulation can be effective in improving recovery. Other differences include mechanistic explanations behind the therapeutic effects of ChR2 stimulation. For example, Cheng et al.[28] reported that ChR2 stimulation led to increased cerebral blood flow and expression of neurotrophins in the motor cortex contralateral to the stroke. Our data show that stimulating thalamocortical axons directly impacts their wiring, but did not induce significant changes to regional cerebral blood flow. The reason for these differential effects on blood flow can be explained by the fact that the previous study[28] directly and continuously stimulated cortical neurons at 10 Hz for 1 min epochs (600 pulses per minute), which we independently confirmed (see 'Results' section), whereas our study employed a lower intensity regime with intermittent 5 Hz stimulation (60 pulses per minute). Regardless of what direct impact optogenetic stimulation had on cerebral blood flow, our study (using IOS imaging) and the previous one both report enhanced neurovascular coupling with optogenetic therapy. Whether our stimulation parameters could have impacted the expression of trophic factors in the contralateral hemisphere is unknown, but would seem less likely since we induced a small focal cerebral infarct, which tends to limit involvement of the undamaged hemisphere[59].

The ability to remotely activate and target specific cell types or circuits through optogenetics is a primary advantage over more general brain stimulation methodologies, such as transcranial magnetic stimulation (TMS) or transcranial direct current stimulation (tDCS), which can excite all cell types and thereby produce unwanted side effects. In our study, we did not observe any obvious deleterious side effects such as seizure activity, weight loss (Supplementary Fig. 8) or maladaptive behaviours during the 1 h stimulation period or afterwards. Although our study represents an early step in understanding the potential for optogenetic approaches to rewire stroke-affected circuits and improve outcome, much more work is needed. For example, future studies should assess whether the pairing of optical stimulation with specific movements during rehabilitation can further enhance limb recovery. This approach was successfully employed in a previous study by pairing electrical stimulation of the ventral tegmental area with forepaw reaching in rats recovering from cortical lesions[60]. It would also be worthwhile to electrophysiologically probe the effects of optogenetic stimulation on cortical and thalamic neuronal activity levels[61], which influence sensory processing, attention and recovery from damage. Another consideration for future work is to test whether red shifted opsins, which allow for deeper penetration/activation in the brain, can be used to improve stroke recovery in larger animals. And finally, while we recognize the need for much more work at the preclinical stage, the efficacy it has shown in this study and the fact that optogenetic therapies are now being tested in human clinical trials, suggest there is reason for cautious optimism.

## Methods

**Animals.** Two- to five-month-old male C57BL/6 mice were used in the present study. Mice were socially housed in groups of 2–4 under a 12 h light/dark cycle and given *ad libitum* access to water and standard laboratory diet. All experiments were conducted according to the guidelines set by the Canadian Council for Animal Care and approved by the University of Victoria Animal Care Committee.

**Thalamic injections and cranial window surgery.** Mice were anaesthetized with isoflurane (2% induction; 1.5% maintenance) in medical air and fitted into a custom-made stereotax. Body temperature was maintained at 37 °C using a rectal thermoprobe and a temperature feedback regulator. A 0.03 ml bolus of 2% dexamethasone was given subcutaneously to reduce inflammation during and after the procedure. A thin layer of cyanoacrylate glue was applied to the exposed, dry skull and a custom-made magnetic metal ring (12 mm outer diameter, 6 mm inner diameter) was adhered to the skull overlying the right somatosensory cortex. Using a high-speed dental drill, a burr hole was drilled at 1.6–1.8 mm posterior and 1.8 mm lateral to bregma to position injections in the ventral thalamus (see Supplementary Fig. 1). A 33-gauge Hamilton syringe was lowered 3.5 mm ventral to the cortical surface and slowly injected 0.5 μl of HEPES buffered artificial cerebrospinal fluid (ACSF) solution containing AAVs purchased from Vector labs (AAV2.CAG.eGFP) or the University of Pennsylvania Vector Core: AAV2.CAG.eGFP (1:50 dilution), AAV1.CB7.CI.mCherry.WPRE.rBG (1:100 dilution), AAV2.CaMKII.hChR2(E123A).eYFP.WPRE.hGH (1:10 dilution), AAV1.Syn.GCaMP6s.WPRE.SV40 (1:10 dilution).

Following AAV injection, a cranial window was installed as previously described[62]. A 4 mm diameter craniotomy (incorporating the burr hole) was drilled in the centre of the metal ring, leaving a 1–2 mm border of skull inside the ring. Cold HEPES-buffered ACSF was intermittently applied to the skull during the drilling procedure to keep the brain cool. The bone flap was removed, leaving dura intact, and the brain surface was kept moist with cold ACSF. A 5 mm circular coverslip (no. 1 thickness) was placed over the brain and secured to the skull and magnetic ring with cyanoacrylate glue and dental cement. Mice were allowed to fully recover from anaesthesia under a heat lamp before returning to their home cage. Mice that showed any loss of clarity to the imaging window before the stroke were excluded from the study.

**Targeted stroke in forelimb somatosensory cortex.** A photothrombotic stroke[63] was induced through the cranial window 1 day after the last BL imaging session. Mice were anaesthetized with isoflurane and placed under an Olympus BX51WI microscope. A green LED (532 nm, 25.8 mW at the back aperture) was focused onto the cortical surface through a ×10 objective. Green light was targeted to a ~1 mm diameter area centred on the forelimb somatosensory cortex. Mice were

injected with 1% rose bengal solution (100 mg kg$^{-1}$ in HEPES-buffered ACSF, i.p.) and the targeted area was illuminated for ~18 min until vessels stopped flowing. Sham stroke control mice were injected with rose bengal solution but not exposed to green light.

**Calcium imaging and analysis.** Mice that had been injected with AAVs to drive GCaMP6s and mCherry expression in thalamic neurons were implanted with a cranial window. BL imaging sessions began 4 weeks after cranial window surgeries since previous studies have shown that any surgery-induced inflammation and gliosis has subsided by 3–4 weeks recovery[64]. Mice were lightly anaesthetized with 0.9–1% isoflurane in medical air for each imaging session and placed under an Olympus FV1000MPE multiphoton laser scanning microscope equipped with a mode-locked Ti:Sapphire laser tuned to 770 or 940 nm for excitation of mCherry or GCaMP6s, respectively. Laser power delivered by each wavelength ranged from 20 to 55 mW at the back aperture depending on imaging depth. Images were acquired through a ×40 Olympus IR-LUMPlanFl water-immersion objective (NA = 0.8), using Olympus Fluoview FV10-ASW software. The dorsal surface of the contralateral forepaw was stimulated with a pencil lead (attached parallel to the middle digits) connected to a piezoelectric bending actuator (Piezo Systems, Q220-A4-203YB; ~300 μm deflection) controlled by an isolated pulse stimulator that delivered vibrotactile stimulation for 1.5 s at 100 Hz stimulus (5 ms pulse width and 10 ms interpulse interval). The loss of dark fur on the forepaw allowed us to reposition the pencil lead to the same position each week. Only one stimulation placement was used for each imaging session. Thalamocortical axons in layer 1 and 2 (~20–120 μm below cortical surface) and within 400 μm of the infarct border, which we denote as 'peri-infarct', were selected for imaging. All efforts were made to follow axon segments within the same region of cortex across imaging sessions. Images (512 × 512 pixels, 0.248 μm per pixel) were collected at 4 Hz for 5 s before and after the start of forepaw stimulation for each trial, with a total of 6–8 trials per imaging session.

For analysis, individual trials were corrected for movement using automated plug-ins in ImageJ software (StackReg and TurboReg)[65]. Trials were then averaged and GCaMP6s-expressing thalamocortical boutons were identified from an average intensity projection from all 40 frames. The fluorescence time-course was measured with a circular ROI (area = 1.73 μm$^2$) placed over the centre of each bouton, which averaged all pixels within. Changes in fluorescence relative to BL fluorescence was calculated as $(F - F_0)/F_0$ with $F_0$ as the median fluorescence over 5 s before forepaw stimulation. Thalamocortical boutons were considered forepaw responsive if $\Delta F/F_0$ for the 2.5 s after stimulation was significantly greater than values 2.5 s before stimulation (two tailed *t*-test assuming unequal variance). Data were exported to Clampfit 9.0 software for analysis of peak amplitude, time to peak and half-width of responses to forepaw stimulation.

**Imaging thalamocortical structural dynamics.** Mice were anaesthetized with ketamine (100 mg kg$^{-1}$) and xylazine (20 mg kg$^{-1}$). This dose reliably provided 60–90 min of anaesthesia and sufficiently reduced breathing artifact during imaging. High-resolution *in vivo* two-photon images were acquired with a ×40 objective (NA = 0.8), using Olympus Fluoview FV10-ASW software. Fluorophores were excited with the laser tuned to 770 nm for mCherry and 900 nm for GFP or YFP excitation (power at back aperture between 17 and 49 mW). Three imaging areas per mouse were selected based on proximity to the FLS1 cortex (the prospective infarct location), such that the areas would be located within 400 μm of the putative infarct border. These peri-infarct regions were on average 153 μm from the infarct border with a standard deviation of ± 88 μm. There were no significant differences between the two stroke groups in imaging distance from the infarct border (P = 0.47 based on t-test). In each area, fluorescently labelled axons were imaged to a depth of ~150 μm below the pial surface. Image stacks were collected at 1.25 μm z-steps covering an area of 151 × 151 μm (800 × 800 pixels, 0.189 μm per pixel), averaging three images per section. Animals were imaged twice before stroke and once weekly for 5 weeks after the stroke or sham procedure, after which window clarity tended to degrade.

Sixteen-bit stacks of images were imported into ImageJ or FIJI software and median filtered. All analyses were conducted by an experimenter blinded to experimental conditions. Changes in axon density were assessed by measuring the area fraction of fluorescent axon segments within each imaging area over time. For analysis of synaptic boutons, thalamocortical axons were traced in three dimensions and the intensity of fluorescence through the centre of the axon shaft was plotted over distance. EPBs were defined as bright swellings along the axon backbone that exceeded twice the median fluorescence of the shaft[34], whereas TBs were defined as projections off the axon backbone that ranged in length from 1–5 μm (ref. 33). Turnover of boutons was determined by comparing images from two consecutive weeks and determining whether a given bouton was located in the same position in both weeks ('stable'), or had disappeared or formed in the second week. Turnover was calculated as the sum of lost and gained boutons divided by the sum of the total number of boutons counted for both time points. Turnover was then further broken down into fraction eliminated and formed by dividing either the boutons lost or gained by the summed total at both time points. Every effort was made to analyse the same axon segments in each animal throughout the entire study. In cases when an axon segment was lost or obscured, measurements

were collected from a nearby axon with similar characteristics for the remainder of the study.

**Laser speckle contrast imaging.** The cortical surface was illuminated with a 785 nm elliptical laser beam (2.4 × 3.4 mm) coupled to a 3 × beam expander (ThorLabs; 1–3 mW output power) in lightly anaesthetized mice. Twelve-bit images were collected with QCapture software from a CCD camera mounted on an upright Olympus microscope through a × 4 objective. Each imaging session involved collecting 100 image frames (10 ms exposure time) immediately before optogenetic or control stimulation (time 0) and then at 20, 40 and 60 min intervals during the stimulation protocol. At each of these time points, the LED cap was quickly removed for 60–90 s to collect laser speckle data. For positive control experiments, laser speckle images for blood flow were collected immediately after delivering 6% $CO_2$ for 90 s or 10 Hz (20 ms pulses) continuous optogenetic stimulation for 60 s. As previously described[66,67], the laser speckle contrast factor (between 0 and 1) was calculated based on the ratio of the standard deviation to the mean intensity in a small region (3 × 3 pixels). Values closer to 0 indicate higher blood flow due to blurring of speckles from moving objects in vessels. For quantification, the speckle contrast values were measured in ROIs (0.5 × 0.5 mm region) avoiding large surface vessels and expressed as a percent change relative to BL.

**IOS imaging.** IOS imaging was conducted on a weekly basis before and after stroke, as previously described[7]. Mice anaesthetized with 1% isoflurane were placed under an upright microscope. Using a × 2 objective, the vasculature on the cortical surface was visualized by collecting reflected white light filtered through a YFP emission filter (523–550 nm). The depth of focus was set just below the brain's surface to lessen the contribution of large surface vessels to measured haemodynamic responses. For imaging intrinsic signals, the cortical surface was illuminated with a red LED (635 nm) driven by a 1 A current driver. The contralateral fore- or hind-paw was stimulated at 100 Hz for 1 s using a pencil lead connected to a piezoelectric bending actuator (as described above in the calcium imaging section). Each session consisted of 12 stimulation/no stimulation trials with a 10 s interval between each trial. For individual trials, images were collected over 3 s at 30 Hz, with 30 ms exposure using a MiCAM02 camera (SciMedia) for 1 s before limb stimulation and for 2 s after. The percent change in light reflectance as result of stimulation ($\Delta R/R_o$) was calculated by first mean filtering images (radius = 5 pixels), followed by subtracting and then dividing each image frame from an average pre-stimulation image. IOS maps were analysed for the peak change in reflectance based on a 0.5 mm diameter region of interest placed in the centre of the cortical response. The area of cortical maps was based on a half-maximal response threshold.

**Optogenetic therapy.** Stimulation was delivered by a blue XPE Series Cree LED (475 nm, super bright LEDs) connected to a 1A LED current driver. A doughnut-shaped neodymium magnet with an external diameter of 12 mm and an internal diameter of 6 mm was glued to the LED. Stimulation consisted of 5 ms light pulses at 5 Hz for 1 s, with a 4 s OFF period between trains. Mice were randomly selected to receive 1 h of optogenetic stimulation sessions beginning 3 days after stroke (always a Monday) and continued for five sessions per week for up to 6 weeks, or control stimulation where blue light was given in the absence of ChR2 expression or ChR2 expression without blue light. The time of day that mice were given stimulation (afternoon) was kept consistent across days. Light weight flexible wires connecting the headcap to the LED driver, allowed mice to freely move around the cage.

**Field potential recordings.** To validate optogenetic activation of ChR2-expressing thalamocortical axons in vivo, isoflurane-anaesthetized mice underwent a craniotomy to expose the somatosensory cortex. We used a 473 nm DPSS laser to assess cortical responses to different power densities, including one that matched our 10 mw mm$^{-2}$ head-mounted LED. Field potentials were recorded by inserting a 1–2 MΩ glass micropipette filled with HEPES-buffered ACSF at a 30° angle 200 μm below the cortical surface in the illuminated area. Single 5 ms light pulses or 5 Hz trains were delivered at 2.5, 5, 10 and 40 mW mm$^{-2}$ light intensities. Evoked potentials were amplified (1,000 ×) and filtered between 1 and 1,000 Hz with a differential amplifier (A-M Systems). Forepaw (5 ms deflection) or light-evoked cortical responses were recorded every 10 s and averaged over 30–40 trials. Following stimulation experiments, synaptically driven cortical depolarizations were blocked with surface application of the ionotropic glutamate receptor antagonists CNQX and AP5 (0.5 and 1 mM, respectively).

**Ladder walking and tape removal tests.** Mice were assessed on the ladder walking and tape removal tests[36,68] over 3 weeks before stroke to establish a BL level of performance and once weekly afterwards. For each session of testing, mice underwent three trials per task. For the ladder walking test, each trial consisted of one crossing of a horizontal ladder (70 cm long) with unevenly spaced rungs (1–2 cm spaces, 1 mm diameter rungs), while videotaped from below. Steps were visualized using slow-motion video playback and scored by an observer blind to

condition as one of three categories: slip, partial or correct placement. A correct step occurred when the mouse placed its paw centred on the rung, so that the weight was supported by the paw. A partial step was one where the paw partially grasped the rung so that either the heel or the toes were touching the rung. A slip was categorized as a step that either completely missed or a step that touched the rung but then slid off, causing a slight fall. The fraction of each type of step was estimated by dividing by the total number of steps.

For the tape removal test, each trial was initiated by placing 5 mm circular pieces of tape onto the palmar surface of both forepaws and placing the mouse into a clear glass beaker, filmed from below. The animal was allowed to remove the pieces of tape with the teeth, to a maximum of two minutes of testing time. A blind observer recorded latencies to remove each piece of tape using slow-motion video playback. An asymmetry score was calculated by dividing the latency of the impaired (left) paw by the total latency summed across both paws. A score of 0.5 indicated no asymmetry, while a score greater than 0.5 indicated a longer latency to remove the tape from the impaired paw.

**Histology and measurement of infarct volume.** A subset of mice ($n = 6$ mice per group) were killed after 10 days of stimulation or control stimulation with an overdose of pentobarbital (175 mg kg$^{-1}$, i.p.) and perfused with 0.1 M phosphate buffer followed by 4% paraformaldehyde. Brains were post-fixed at 4 °C and sliced into 50 μm-thick coronal sections using a Leica VT1000 vibratome (Leica Biosystems). Every third section was mounted onto charged slides and stained with cresyl violet. An experimenter blind to condition traced the infarct zone using ImageJ software. Infarct area was measured in each section and multiplied by the distance between sections to estimate infarct volume.

**Statistics.** Sample sizes were based on comparable $n$ values used for similar experiments in the literature. For most data sets, a two-way repeated-measures analyses of variance (ANOVA) was calculated in SPSS software to identify significant differences between groups, over time and group by time interactions. In cases where groups were compared at the same time point, either a two-sample $t$-test or one-way ANOVA was used, depending on the number of conditions. If the ANOVA revealed a significant main effect, post hoc $t$-tests were used to further identify group differences at specific time points or differences post-stroke relative to BL. $P < 0.05$ was considered significant for all tests. # indicates significant comparisons within a group (for example, BL versus 1 week), whereas * indicates significant comparisons between groups (for example, control stimulated versus stimulated). Data presented in graphs are mean ± s.e.m.

**Data availability.** The data that support the findings of this study are available from the corresponding author on reasonable request.

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

## Acknowledgements

The authors thank Kerry Delaney, Gautam Awatramani, Patrick Reeson and Eslam Mehina for technical advice and critical feedback on the manuscript. We are grateful to the scientists of the Genetically-Encoded Neuronal Indicator and Effector (GENIE) Project at the Janelia Farm Research Campus of the Howard Hughes Medical Institute and Dr Karl Deisseroth of Stanford University for generously allowing GCaMP6 and ChR2 viral constructs to be distributed through the Penn State Vector Core. This work was supported by operating, salary and equipment grants to C.E.B. from the Canadian Institutes of Health Research (CIHR), Heart and Stroke Foundation (HSF), Michael Smith Foundation for Health Research (MSFHR) and Natural Sciences and Engineering Research Council of Canada (NSERC). K.A.T. was supported by postdoctoral fellowships from the CIHR and the Canadian Diabetes Association.

## Author contributions

K.A.T.: experimental planning, analysis and execution; writing and editing of manuscript. S.L.T. and E.R.W.: execution of experiments. C.E.B.: experimental planning, execution and analysis; writing and editing of manuscript.

## Additional information

**Competing interests:** The authors declare no competing financial interests.

