## [Peer Review File · Nature Communications]

Reviewers' comments:

Reviewer #1 (Remarks to the Author):

This manuscript by Tennant and colleagues investigates how intermittent optogenetic stimulation of thalamocortical (TC) axons in superficial layers of somatosensory cortex around the infarct might enhance TC bouton turnover and augment functional recovery after focal stroke in mice. The study has several strengths. First, the authors use advanced experimental techniques, such as two-photon microscopy, intrinsic signal imaging and optogenetics in vivo. These are not trivial experiments. By tracking the dynamics of axon boutons (and their responses to sensory stimulation) before and after stroke in the same animals, they are able to document clear changes in both structure and function of TC axons as a result of stroke and optogenetic stimulation with a relatively small sample size. Second, they present some new and interesting results: 1. They report a substantial decrease in the fraction of axons that respond to forelimb stimulation in peri-infarct cortex after stroke (from 24% to 10-15%; Fig. 2C); 2. Increases in both TB and EPB turnover after optogenetic stimulation of TC axons; 3. An enhanced recovery of the forelimb sensory map in stroke animals after chronic optogenetic stimulation of TC axons; and 4. A clear effect of TC axon stimulation on enhancing functional improvement after stroke (without a change in infarct volume). Overall, the findings presented are convincing, the figures are well crafted (except perhaps for Fig. 2, see below), and the quality of their imaging data is very good. I agree that their approach of chronic stimulation for weeks stands out as unique compared to previous papers of optogenetics in stroke.

But there are also some weaknesses that limit the potential significance of this study. The limitation that stands out the most is the lack of important control experiments. There is no smoking gun showing that changes induced by optogenetic stimulation are related to stroke or to the restoration of axonal bouton responsiveness (see below). As a result, I am not enthusiastic about the publication of this study in its present form. However, it should be possible for the authors to perform appropriate controls that would mitigate these concerns.

1. All Figures: It seems critical to show the targeting of the virus injections in multiple mice. They show a one panel for one animal in Fig. 1b. but it would be important to confirm the correct targeting of rAAV expression in thalamus in the majority of the animals. Providing a summary diagram of the outlines of the expression areas in VPL for these animals would be very useful.

2. Figure 2: The data in this figure is the basis for their rationale to use optogenetics – they find a sustained decrease in the fraction of TC boutons responsive to FL stimulation after stroke. But they do not show that the fraction of FL-responsive TC axon boutons is stable in control animals after a sham stroke (Rose Bengal without light, or better yet, light without Rose Bengal). This is important because it is conceivable that fluorescence signal drop out could have occurred due to changes in window quality, GCaMP6s expression or toxicity, due to inflammation/edema as a result of the stroke, from photodamage, etc. There is also likely a significant decrease in the density of boutons after stroke, but this is never quantified (see

point #4 below); so the decrease in % FL-responsive boutons could be due in part to loss of boutons. For specificity, it would also seem important to show that there are no changes in the fraction of boutons that respond to hindlimb stimulation after stroke in FLS1, but this control is also missing.

In Fig. 2b (top row), they should show the loss/shrinkage of the FLS1 map after stroke and its partial recovery at 4 weeks, as well as the HL map (presumably stable) throughout. Overlaying the baseline maps is not so helpful considering the tissue distortion that occurs after the involution of the infarcted tissue post-stroke.

For Fig. 2d, the text states they saw no differences in response amplitude and latency of bouton calcium responses after stroke, but the representative cells shown in the figure have a much lower amplitude (and slightly longer latency). The authors should show graphs for latency and amplitude that include data for individual animals.

3. Figure 3g/h: Instead of the 6% CO₂ control, it would be better to show laser speckle images of changes in blood flow after stroke. They should also show the data for 10 Hz stimulation (17% increase in blood flow). How was the data collected at 20 min intervals anyway if the mice are wearing the magnetic headcap with LEDs for the entire 60 min? Is it possible that with anesthesia they can't see an effect of optogenetics on blood flow?

4. Figures 4-7: An important control that is missing is the use of optogenetics in mice with sham stroke (plus perhaps also image axons in the HLS1 region). The goal is to make sure that optogenetic stimulation is not having a non-specific effect on TC axons that is independent of stroke. Earlier, the authors made a point to say that stroke dampens the excitability of TC boutons, so restoring that excitability would help stroke recovery. Then, they must show that in normal mice, ChR2 stimulation of TC boutons has little effect on bouton turnover. Of course this sham stroke control is also important for intrinsic signal imaging (Fig. 6) and behavioral experiments (Fig. 7). Moreover, they should show that optogenetics restores to normal the fraction of axon boutons that respond to FL stimulation in peri-infarct cortex?

In Fig. 4d, does stroke (or optogenetics) change the relative proportion of TB and EPB boutons?

Since the rates of bouton formation and elimination are different after stroke and to different degrees (esp. for EPBs), one would expect that bouton density would be higher after optogenetics. What happens to the density of EPBs and TBs? Also, since the density of boutons likely changes a lot after stroke (huge increase in bouton elimination rates but flat rates of formation), wouldn't it be more accurate to measure elimination/formation rates as a function of axon length (per micrometer)?

They do mention a sham stroke control for looking at long-term EPB and TB turnover, but the sample size was small (n= 3) and they don't show the data. It would be important to add those data to Fig. 4f-h and 5b, c, e.

Other comments:

- One of the most surprising for me was the persistent decrease in TC axon density in peri-infarct cortex for >1 month after stroke (Fig. 1d). I would have expected the opposite; since the thalamus is still receiving inputs from the periphery, I imagined that sprouting of new TC axons would be a good mechanism for structural plasticity to mediate remapping of list functionalities. Perhaps the authors can comment on this and venture an opinion as to how TC axons might mediate the stronger map size/signal for FL stimulation after optogenetic stimulation. Was there a significant correlation between map size/signal and axon turnover?
- Is the partial recovery of the % axon boutons that respond to FL stim significant (Fig. 2c, top of page 4)? This also applies to graphs in Figs. 4f-g, 5b-c, 6b/d and 7a-b: there are always stats on the differences between control and stimulated, but no stats for the effects of stroke
- How do the authors distinguish between the appearance of new boutons vs. the enlargement of pre-existing small boutons (p. 4; Fig. 2e, bouton #4)? Was it always based on mCherry expression and multiple z-slices were checked (instead of deciding based on GCaMP6s expression in a max projection)? What % of newly formed boutons were responsive to FL-stim?
- Fig. 2e gray bars are sometimes in foreground, sometimes not.
- “such as beading of cortical dendrites” (p. 5). How do they see dendrites when only axons are labeled? Actually, it would be interesting to see if dendrites are more sensitive to LEDs. (just as they are more sensitive to photodamage and to excitotoxicity). Just curious if they have looked.
- “...but did NOT induce changes in regional blood flow” (p. 6).
- The rates of bouton turnover seem a bit higher than those reported by de Paola (2006). The survival fraction of type A1 EPBs reported by de Paola was ~80% after 3-4 weeks. In this study, it's more like 50% in 5 weeks.
- Fig. 6a: They should add guides to orient the reader as to where rostral-caudal and medial-lateral directions are.
- “our estimate of TC bouton responsiveness... is in close agreement with previous work...” (p. 10). This warrants some clarification. How exactly are they comparing chronic calcium imaging of boutons from thalamic neurons using GCaMP6s to acute calcium imaging neuronal somata of L2/3 cortical neurons using OGB? The statement should probably be removed.
- The data on Fig. 7d are underpowered (n= 4) to make a reliable conclusion, and the experimental time point for estimating infarct volume does not match the rest of the study

(10 days vs. 10 weeks!). Either repeat these infarct volume measurements at 10 weeks post-optogenetics (with more mice) or remove these data.

- To reduce unnecessary clutter in the main figures, consider moving the following panels to suppl data: 2b (blue panels); 3c; 4d (left diagram: just label the EPB and TB in panel e);
- The should show the different control stimulation protocols (with Chr2 but no light, no Chr2 plus light) in suppl. Data.

Reviewer #2 (Remarks to the Author):

This study teases out the relationship between thalamocortical connections to the forelimb of the somatosensory cortex and an ischemic injury by stroke. The paper argues that optogenetic stimulation of the axons of these connections, which shares characteristics of vibrotactile stimulation at the same frequency, can enhance behavioral recovery. The study argues that this enhancement is due to enhanced turnover of "boutons" on the axons, not due to axonal sprouting or increased dendritic branching. In addition to the increased turnover of thalamocortical boutons, they distinguish between en-passant boutons (EPB) which show an increased turnover in the earlier phases of stroke recovery [2-3 weeks], and terminaux boutons (TB) which show increased turnover in the later phases of stroke recovery [3-5 weeks].

Firstly, the study shows that thalamocortical axons are less responsive to vibrotactile stimuli after stroke, as visualized by GCaMP6s signaling. Secondly, optogenetic stimulation of axons does not increase cerebral blood flow, as studied by laser speckle contrast maps. Thirdly, (as stated above) optogenetic stimulation promoted enhanced thalamocortical axon EPBs and TBs turnover in the 2-3 weeks and 3-5 weeks following stroke respectively, as opposed to controls receiving Chr2 without light, or light without Chr2. Axonal branching was noted not to differ. Lastly, behavioral recovery in horizontal ladder walking and adhesive tape removal following stroke was also enhanced in the optogenetic group. Improvements persisted beyond the optogenetic stimulation and was significantly correlated to EPB formation in weeks 2-3.

The manuscript is novel in that it studied thalamocortical circuitry for somatomotor stroke recovery, rather than cortical neurons that are more commonplace in the literature. Additionally, one of the study's measured output was boutons, rather than larger changes such as axonal branching, allowing smaller changes in structure to be analyzed. The optogenetic stimulation was also applied for much longer, 6 weeks, then other studies which stopped after 4-7 days. One nice control is the results on cerebral blood flow, showing that the improvements observed were not merely due to increased blood flow, which was also shown to be possible with higher frequency of optogenetic stimulation.

Major Criticisms

- 1) The discussion of EPB vs TB functions argues that EPBs turnover was involved in initiating

the behavioral recovery, while TBs may be involved in maintaining this improvement. While the correlation in the results between EPB and enhanced motor recovery supports the first claim, there was no evidence to support the later claim; for example, an experiment stopping the optogenetic stimulation after 2-3 weeks and showing that TB turnover is not enhanced and that the behavioral recovery would not be retained in the long term would elegantly prove this claim. However, if an experiment like this cannot be completed, perhaps a sentence regarding that the function of the TBs remains a mystery.

2) Fig 6, the increase in cortical response follow stroke is described as "remapping." in Fig 2, no increase in cortical response was found during recovery period and this discrepancy is not fully discussed. Is the conclusion from IOS results that over recovery the BOLD response becomes stronger independent of activity levels?

3) The statistically equivalent ChR2- and ChR2+ groups (Fig. 5e) suggest that optical illumination itself may be a confounding factor. It would be important to show ChR2- results throughout the study for all measures since stimulation through ChR2 (not indirect effects of warming due to laser light) is the main point of the study.

Minor Criticisms

1) In the first section of the results, it is stated that before stroke 25% of GCaMP expressing boutons responded to vibrotactile stimulation and that this number drops significantly after stroke. Please include the percentage that it drops down to in the text (top of page 2) because it would read better.

2) Figure 2c states that boutons recover, but in reality- the last measure (4 weeks post stroke) is not significantly different from the first... although the trend is there.

3) Figure 2b- the images in blue, upon visual inspection, don't seem to show that the boutons are recovering from 2 weeks to 4 weeks.

4) Figure 5d: the blue shading is not defined in the legend (one assumes that a "stable" bouton is as in figure 5a), and the pre-existing bouton that is removed in the +5 weeks' line, should be colored in red shading (and included in the legend as "eliminated") at the +1-2-week line.

5) It is confusing why Figure 5e shows that there pre-existing boutons at control and stimulated did not have significantly different survival at week 5, while figure 5c shows a significant difference of EPB elimination at week 4.

6) Figure 6 is described as "remapping." While this may be true no mapping was actually done, therefore it is unclear if recovery of responses returned in the same topographically organized manner as before. Reword these findings as "functional recovery" of cortex rather than remapping.

Reviewer #3 (Remarks to the Author):

This paper shows that trans-cranial optogenetic stimulation of thalamocortical axons (1 hour daily session 5Hz trains; 1 sec on, 4 sec off) is effective in promoting the formation and stabilization of thalamocortical synaptic boutons, the re-emergence of forelimb derived sensory cortical maps, and recovery of sensorimotor abilities. The results are interesting and fairly novel. Previous work has used stimulation of specific brain sites to aid in functional recovery. Some of these studies are cited, but a relevant set of studies that used "contingent" electrical stimulation of the VTA to facilitate functional recovery after FL motor cortex lesion is missing and seems relevant (e.g. 8577374).

1. The present study employed relatively small infarcts in FLS1. Figure 2 uses GCaMP6s to show that thalamocortical axons are less responsive after stroke. The authors measured calcium responses in axonal boutons before and after the infarct. The imaging portion of this experiment seems straight forward since the brain can be imaged during every session and major features used for alignment. However, what is not at all clear is how they stimulate the forepaw in every session in the same reproducible way. It is possible that the results shown reflect differences in the placement of the stimulation device. What was done to assure that the placement was identical in each session? Note that this is as critical as aligning the imaging part during each session. Ideally, one would use an evoked response that does not change as a result of the manipulation (infarct) and serves to place the paw stimulator. Otherwise, how can we be sure that the stimulus is identical between sessions? Many details are missing about how the stimulation was delivered. For example, was the stimulator only placed in one spot per session? Interpretation of these results and those in figure 6 relies as much on the imaging part as on the stimulus configuration.

2. One of the main findings of the study is that bouton TOR is affected by both stroke and optogenetic therapy. It would be useful to show the effects of the same optogenetic therapy in control non-infarcted mice. This seems relevant because it would clarify if the effect observed is just a natural feature of optogenetic stimulation. It would also reveal if normal FLS1 representations (IOS) are affected by this stimulation. In other words, a non-lesioned set of controls (with and without optogenetic stimulation) undergoing the same imaging procedures (IOS and 2P) would be useful. If this is not supplied, the authors should discuss the reasoning for the lack of these relevant groups.

3. Related to the previous points, the authors did not seem to attempt to test the effect of optogenetic therapy on either calcium or field potential sensory-evoked responses. Was this attempted? What were the results?

4. Figure4 shows a timeline of experiments. Were the IOS, 2P and behavior done in the same animals? If so, this appears to indicate that animals underwent weekly sessions of isoflurane and ketamine/xylazine anesthesia. This needs to be clarified. If it was the same animals, a description of the timeline of procedures per week should be described.

5. The other major finding of the paper is that functional recovery occurs at the behavioral level after optogenetic therapy. It is rather mysterious why a synchronous activation of

thalamocortical fibers at 5 Hz without any relation to behavior would have a beneficial effect on behavior. The effects on boutons seems plausible, but why would such a crude form of stimulation without any relation to ongoing behavior have a beneficial effect? For example, in the functional recovery study noted above, the electrical stimulation had to be contingent to the correct behavior and rewarding. But in this case, the stimulation lacks any relation to ongoing behavior. A different interpretation is that optogenetic stimulation makes the animals more active/alert (or simply "activates" the S1 cortex) and therefore leads to better behavioral outcomes in the tasks. Indeed, the sensory thalamus is known to control the state of cortical activation (20053845), which controls incoming sensory inputs. Such state-related changes would be distinct from those related to circuit rewiring necessary for sensorimotor skills recovery. A discussion of these important issues would be useful.

Reviewer #4 (Remarks to the Author):

This is a sophisticated analysis of neural plasticity in the context of recovery after ischemic stroke. Overall, the results are impressive and the experiments are well-designed. Major findings are that stroke reduces number of thalamic axons visible in L1, that there terminal boutons are particularly destabilized, gCaMP imaging in thalamic boutons shows that activity is suppressed in a long term way. The authors then go on to show that repeated optogenetic stimulation of thalamic inputs to L1 can promote functional recovery, using intrinsic signal imaging as well as behavioral measures. The data as presented are beautiful, and the methods are state-of-the-art. However, there are a few gaps in the logic of the paper that would significantly enhance its impact. The effects of optogenetic stimulation on thalamic boutons is well-characterized – and somewhat subtle, making it hard to appreciate the influence altered bouton dynamics will have on thalamocortical transmission – but the effects on gCaMP6 signals is not evaluated. As it stands, the anatomical imaging of boutons is rather anecdotal, and we cannot attribute the change in intrinsic signaling imaging nor the behavioral recovery to a particular process – thalamic transmission or neocortical circuit rearrangement.

1. Control experiments are stroke without stimulation. How dynamic are axons under baseline conditions? If you stimulate axons in the absence of stroke, do you get remodeling? This is important to evaluate the application of the technique to functional enhancement over multiple conditions. Also, because the stroke area can be hard to assess and the window of analysis can vary accordingly, it is important to understand how location of stimulation wrt whether it is closer to normal tissue or in the infarct one related to the degree of plasticity.
2. It is interesting that the investigator makes the tacit assumption that the thalamocortical fibers they are observing must come from the VPM. In fact, thalamic axons in L1 are likely to come from a higher-order thalamic nucleus, such as Pom. This then raises the question of whether VPM axons in L4 and L5b have the same properties as Pom axons.
3. The authors nicely compare the effects of forelimb stimulation to optogenetic activation of thalamic fibers, which have many similarities. However, optogenetic stimulation induces a response with multiple components – probably inhibition, since the synchronized stimulation

with light is likely to activate some different intracortical pathways. This difference may be critical in dissecting therapeutic options for thalamic activation in stroke recovery models. How was the light intensity for optogenetic stimulation set – so that it matched the forelimb stim field potential? Or, was it maxed out?

4. In Figure 2, the authors show that thalamic axons are less responsive to forelimb stimulation for weeks after stroke, using signals from gCaMP6. Intrinsic signal imaging in Figure 6 indicates that optogenetic stimulation can induce a change in blood flow. However, the missing piece here is that thalamic axons are becoming more responsive, with stimulation. The anatomical data is all rather descriptive in this regard – it is hard to determine how altered dynamics of EPB or TB will change stimulus-triggered activity in the neocortex. For example, we know that there may not be more boutons, but the axons themselves may be more active. An alternate interpretation is that the neocortex is becoming more responsive, but evoked thalamic activity is staying rather constant. Indeed, the notable plasticity of the neocortex might indicate this is the more likely event. The authors should close the loop on these experiments and show which of these scenarios is true, by doing gCaMP6 imaging before and after prolonged optogenetic stimulation.

Minor:

1. Scale looks wrong in Figure 3b (too big) – L1 in mouse is typically ~ 100 μm .
2. Figure 3, but especially Figure 3g,h could be supplemental

Reviewer #1 (Remarks to the Author):

This manuscript by Tennant and colleagues investigates how intermittent optogenetic stimulation of thalamocortical (TC) axons in superficial layers of somatosensory cortex around the infarct might enhance TC bouton turnover and augment functional recovery after focal stroke in mice. The study has several strengths. First, the authors use advanced experimental techniques, such as two-photon microscopy, intrinsic signal imaging and optogenetics in vivo. These are not trivial experiments. By tracking the dynamics of axon boutons (and their responses to sensory stimulation) before and after stroke in the same animals, they are able to document clear changes in both structure and function of TC axons as a result of stroke and optogenetic stimulation with a relatively small sample size. Second, they present some new and interesting results: 1. They report a substantial decrease in the fraction of axons that respond to forelimb stimulation in peri-infarct cortex after stroke (from 24% to 10-15%; Fig. 2C); 2. Increases in both TB and EPB turnover after optogenetic stimulation of TC axons; 3. An enhanced recovery of the forelimb sensory map in stroke animals after chronic optogenetic stimulation of TC axons; and 4. A clear effect of TC axon stimulation on enhancing functional improvement after stroke (without a change in infarct volume). Overall, the findings presented are convincing, the figures are well crafted (except perhaps for Fig. 2, see below), and the quality of their imaging data is very good. I agree that their approach of chronic stimulation for weeks stands out as unique compared to previous papers of optogenetics in stroke.

But there are also some weaknesses that limit the potential significance of this study. The limitation that stands out the most is the lack of important control experiments. There is no smoking gun showing that changes induced by optogenetic stimulation are related to stroke or to the restoration of axonal bouton responsiveness (see below). As a result, I am not enthusiastic about the publication of this study in its present form. However, it should be possible for the authors to perform appropriate controls that would mitigate these concerns.

Q1. It seems critical to show the targeting of the virus injections in multiple mice. They show a one panel for one animal in Fig. 1b. but it would be important to confirm the correct targeting of rAAV expression in thalamus in the majority of the animals. Providing a summary diagram of the outlines of the expression areas in VPL for these animals would be very useful.

Response: As requested, we now provide a summary diagram showing the site of AAV injections in several animals (see Suppl. Fig. 1). Although our micro-injections were directed at the ventral thalamus, the idea of restricting virus expression to just VPL neurons was not possible since the VPL is a small banana shaped nucleus and there are no Cre driver lines that are exclusive to the VPL, at least to our knowledge. Based on our retrograde labelling experiments shown in Figure 1 (after Cholera toxin injection in superficial layers of FLS1 cortex) and reviewing our injection sites in brain sections, we believe that the VPL is a primary source of those axons imaged in FLS1 cortex. With that said, since the PoM is in close proximity to the VPL and also sends axonal projections to the FLS1 cortex (this can also be seen in the Allen Mouse Brain Connectivity Atlas; <http://connectivity.brain-map.org>), it is likely that PoM axons were also imaged in our study. To help mitigate any confusion, we have changed the wording

in the results section of the manuscript to reflect that some axons could have arisen from other thalamic nuclei such as the PoM nucleus, and point out retrograde labelling in the PoM in Figure 1a.

Q2a. Figure 2: The data in this figure is the basis for their rationale to use optogenetics –they find a sustained decrease in the fraction of TC boutons responsive to FL stimulation after stroke. But they do not show that the fraction of FL-responsive TC axon boutons is stable in control animals after a sham stroke (Rose Bengal without light, or better yet, light without Rose Bengal). This is important because it is conceivable that fluorescence signal drop out could have occurred due to changes in window quality, GCaMP6s expression or toxicity, due to inflammation/edema as a result of the stroke, from photodamage, etc.

Response: We completely agree with the reviewer that additional control experiments should show the fraction of responsive boutons in control animals after sham stroke. Therefore we imaged and analyzed TC axonal bouton responsiveness to FL stimulation in 4 mice (1,706 boutons analyzed) that received a sham stroke procedure. Our analysis shows that the fraction of forelimb responsive boutons does not change significantly over the 4 week imaging period after sham stroke. This new data has been added to Figure 2 (black lines). With regard to fluorescence signal drop off due to stroke causing the observed drop in forelimb responsiveness, we have plotted in Figure 2 the normalized fluorescence intensity of GCaMP6s in boutons before and after stroke. This data shows that there is no significant change in GCaMP6s fluorescence intensity over time after stroke.

Q2b. There is also likely a significant decrease in the density of boutons after stroke, but this is never quantified (see point #4 below); so the decrease in % FL-responsive boutons could be due in part to loss of boutons. For specificity, it would also seem important to show that there are no changes in the fraction of boutons that respond to hindlimb stimulation after stroke in FLS1, but this control is also missing.

Response: Our analysis of FL responsive boutons for each time point after stroke was always normalized to the number of thalamocortical boutons present at that particular time point. This was done, as the reviewer notes, to prevent a change in % of responsive boutons being explained simply by a reduction in bouton density. We have modified the text to make it more clear that % responsive boutons was always normalized to the number of boutons present at each week.

Regarding thalamocortical bouton responses to hindlimb stimulation, this is something we considered when we were imaging in the forelimb somatosensory cortex. Although we did not systematically track stroke related changes in the fraction of hindlimb responsive boutons in these areas, we did collect data in a subset of mice primarily before stroke. Based on these experiments, we found relatively little bouton responsiveness to hindlimb stimulation ($4.1 \pm 2.4\%$, 6 mice, 518 boutons) when imaging in the forelimb/peri-infarct cortex region (defined by IOS imaging). This information is now included in the results section related to calcium imaging.

Q2c. In Fig. 2b (top row), they should show the loss/shrinkage of the FLS1 map after stroke and its partial recovery at 4 weeks, as well as the HL map (presumably stable) throughout. Overlaying the

baseline maps is not so helpful considering the tissue distortion that occurs after the involution of the infarcted tissue post-stroke.

Response: This is a fair point, so we have removed the baseline IOS map overlays in the post-stroke images. For GCaMP6s experiments, we performed intrinsic optical signal (IOS) imaging intermittently, such as before stroke and 4-6 weeks later. Therefore we do not have a 1 week post-stroke map to present, although we do have IOS imaging data collected ~4-5 weeks after stroke for the example shown in Figure 2. We now include IOS maps for forelimb and hindlimb at baseline and 4 weeks post-stroke in the Supplementary data section (see Suppl. Figure 3). These maps show the stability of the hindlimb representation/location over time. As for the forelimb map, there was a small piece of the original forelimb representation remaining with diffuse, albeit weaker activation in cortical areas medial to the infarct.

Q2d. For Fig. 2d, the text states they saw no differences in response amplitude and latency of bouton calcium responses after stroke, but the representative cells shown in the figure have a much lower amplitude (and slightly longer latency). The authors should show graphs for latency and amplitude that include data for individual animals.

Response: We agree and have re-assessed our data, particularly the peak response amplitudes of boutons. While there were no significant stroke related changes to mean $\Delta F/F_0$ amplitudes, we did notice that baseline amplitudes tended to be different between animals. Therefore, we normalized peak response amplitudes post-stroke to that obtained before stroke in each mouse and found a significant reduction in amplitude at 1 and 2 weeks post-stroke (vs. baseline, one sample t-tests). Amplitudes returned to relatively normal/baseline levels at 3 and 4 weeks recovery. The mean normalized response amplitudes and latency of bouton responses (as well as individual responses) are now reported in the results section and graphs presented in Suppl. Figure 4.

Q3a. Figure 3g/h: Instead of the 6% CO₂ control, it would be better to show laser speckle images of changes in blood flow after stroke. They should also show the data for 10 Hz stimulation (17% increase in blood flow).

Response: We think the CO₂ challenge experiment is a nice positive control and would prefer to leave this data in the Figure. Our laser speckle experiments examining the impact of 5 Hz intermittent stimulation on blood flow in mice with or without stroke, show little to no change regardless of condition, therefore we think that the example shown in Fig 3g is representative. However, we do agree with the reviewer's suggestion to present laser speckle images for 10Hz continuous stimulation and accompanying graphs. This data is now shown in Figure 3g and h.

Q3b. How was the data collected at 20 min intervals anyway if the mice are wearing the magnetic headcap with LEDs for the entire 60 min?

Response: We apologize for any confusion here. The headcap was removed for ~90 sec of time to collect laser speckle trials at each time point, then immediately put back on the head. This is now clarified in the methods section describing laser speckle contrast imaging.

Q3c. Is it possible that with anesthesia they can't see an effect of optogenetics on blood flow?

Response: Although it is possible that anesthesia could have affected optogenetic changes in blood flow, the fact that we could detect in mice under light anesthesia a 17% increase in CBF with continuous 10Hz stimulation or a 41% increase with brief exposure to CO₂, suggests that our methods should have been sufficiently sensitive.

Q4a. Figures 4-7: An important control that is missing is the use of optogenetics in mice with sham stroke (plus perhaps also image axons in the HLS1 region). The goal is to make sure that optogenetic stimulation is not having a non-specific effect on TC axons that is independent of stroke. Earlier, the authors made a point to say that stroke dampens the excitability of TC boutons, so restoring that excitability would help stroke recovery. Then, they must show that in normal mice, ChR2 stimulation of TC boutons has little effect on bouton turnover. Of course this sham stroke control is also important for intrinsic signal imaging (Fig. 6) and behavioral experiments (Fig. 7).

Response: This is an excellent point, so we ran a completely new set of sham stroke experiments where we compared TC bouton dynamics, IOS responses and behaviour in sham stroke animals that did not receive optogenetic stimulation (n=5 mice) vs. sham stroke animals that received 6 weeks of optogenetic stimulation, 5 days per week, 1 hour per day, beginning 3 days after sham surgery (n=6 mice). Before getting into the details of the results, the simple answer to this big question is that in the absence of stroke (ie. sham stroke) optogenetic stimulation did not have any significant effect on thalamocortical TB and EPB turnover rates, IOS response area or amplitude and behavioural measures of forelimb function (see Suppl. Fig. 7).

More specifically in the absence of stroke, there were no significant effects of stimulation group or time in TB turnover rates (Main effect of stimulation: $p = 0.30$; main effect of time: $p = 0.25$; stimulation x time interaction: $p = 0.44$). Similarly, we found no significant effects of stimulation group or time in EPB turnover rates (Main effect of stimulation: $p = 0.95$; main effect of time: $p = 0.14$; stimulation x time interaction: $p = 0.82$). For IOS responses, we found no significant effects of stimulation group or time in the Peak amplitude of forelimb evoked responses (Main effect of stimulation: $p = 0.69$; main effect of time: $p = 0.44$; stimulation x time interaction: $p = 0.50$) or in IOS Response Area (main effect of stimulation: $p = 0.48$; main effect of time: $p = 0.70$; Stimulation x time interaction: $p = 0.08$). Finally, there was no significant effects of optogenetic stimulation group or Time in the Ladder Walking Test (% Correct steps: main effect of stimulation: $p = 0.14$; main effect of time: $p = 0.48$; stimulation x time interaction: $p = 0.89$; % Partial steps: main effect of stimulation: $p = 0.15$; main effect of time: $p = 0.53$; stimulation x time interaction: $p = 0.95$) or Tape Removal Test (forelimb asymmetry: main effect of stimulation: $p = 0.96$; main effect of time: $p = 0.69$; stimulation x time interaction: $p = 0.98$). Based on these results, we have determined that the beneficial effects of optogenetic stimulation induced by our stimulation protocol are specific to the stroke-affected brain.

Since our sham stroke mice that received chronic optogenetic stimulation were very similar (in all metrics) to those that received control stimulation, we decided to combine these sham stroke data sets together in the main figures so as to reduce visual clutter. However, we have also included all the

ungrouped sham stroke data (plots showing TB/EPB turnover, IOS responses and behaviour in sham stimulated vs no stim mice) in the Suppl. Fig. 7 for readers to evaluate.

Q4b. Moreover, they should show that optogenetics restores to normal the fraction of axon boutons that respond to FL stimulation in peri-infarct cortex?

Response: We would love to do this experiment, but technical limitations make this extremely difficult if not impossible in the context of this revision. Unfortunately as other groups have shown (Rickgauer and Tank, 2009, PNAS; Papagiakoumou et al., 2010, Nat Methods; Prakash et al., 2012, Nat Methods), 2-photon scanning at 940nm (the same wavelength we used to image GCaMP6s) using a 40x objective and comparable laser power as that used in our study, excites and induces photo-currents in ChR2 expressing neurons. This is not surprising since the optimal 2-photon absorption of ChR2 is in the 920-940nm range (~260 GM at 940nm). This unfortunate co-incidence would explain why, to our knowledge, no other group has imaged calcium transients in GCaMP6s expressing neurons that also express ChR2. Recent studies have shown that it could be possible to image GCaMP6 responses in neurons that express a red shifted opsin such as C1V1 (Packet et al., 2015, Nat Methods) or possibly use the blue sensitive ChR2 with a red shifted calcium indicator (RCaMP or JRGEKO). However trying to troubleshoot and implement this novel approach (assuming there was no 2-photon cross talk issues which cannot be guaranteed) would require many months, possibly years to accomplish. Furthermore, we'd then have to redo much of the entire study to prove that all the same optogenetic effects occur with the red shifted opsin. Needless to say, we are definitely interested in this important question and are pushing in that direction, but we think it would be better suited for a future paper.

Q5. In Fig. 4d, does stroke (or optogenetics) change the relative proportion of TB and EPB boutons?

Response: In accordance with the reviewers' suggestion we re-analyzed our TB and EPB bouton proportions and added new data to this analysis. We did not find any effect of stroke (Time) or optogenetic stimulation on the relative proportion of EPB vs TB (Main effect of Group, $p=0.22$; Main effect of Time, $p=0.96$; Group x Time interaction, $p=0.50$). Since there were no significant differences we pooled both groups together and present in Figure 4e the mean EPB vs TB proportions before stroke and up to 5 weeks later.

Q6. Since the rates of bouton formation and elimination are different after stroke and to different degrees (esp. for EPBs), one would expect that bouton density would be higher after optogenetics. What happens to the density of EPBs and TBs? Also, since the density of boutons likely changes a lot after stroke (huge increase in bouton elimination rates but flat rates of formation), wouldn't it be more accurate to measure elimination/formation rates as a function of axon length (per micrometer)?

Response: This is a good suggestion so we now present normalized TB and EPB density data in Figure 4h and 5d, respectively. While the TB density data does not show a clear separation of the stroke groups over time (this is not surprising), our analysis of EPBs reveals a relatively fast and robust restoration of EPB density (to baseline levels) in optogenetically stimulated mice from 2 weeks post-stroke and

onwards that was not evident in control stimulated mice.

Q7. They do mention a sham stroke control for looking at long-term EPB and TB turnover, but the sample size was small (n=3) and they don't show the data. It would be important to add those data to Fig. 4f-h and 5b, c, e.

Response: Indeed the sample size was small so we have added two additional sham stroke mice that received no optogenetic stimulation to reach n=5 mice. We have also added 6 new sham stroke mice that received chronic optogenetic stimulation (n=6). Adding these new sham stroke animals did not change any of our conclusions (see our response to question 4). As recommended we have added sham stroke turnover data to Figure 4 and 5, as well as presented each sham sub-group in Suppl. Figure 7.

Other comments:

Q8a. One of the most surprising for me was the persistent decrease in TC axon density in peri-infarct cortex for >1 month after stroke (Fig. 1d). I would have expected the opposite; since the thalamus is still receiving inputs from the periphery, I imagined that sprouting of new TC axons would be a good mechanism for structural plasticity to mediate remapping of lost functionalities. Perhaps the authors can comment on this and venture an opinion as to how TC axons might mediate the stronger map size/signal for FL stimulation after optogenetic stimulation.

Response: With regard to your question about the persistent decrease in axon density after stroke (Fig. 1d), we were a little surprised as well but the time lapse imaging data clearly indicated that thalamocortical remodelling occurred mostly on surviving branches rather than sprouting of new branches. This finding parallels what several groups (including our own) have described for cortical dendrites after stroke (see Brown et al., 2009-JNsci and 2010-JCBFM, as well as Mostany et al., 2010-JNsci and 2011-JNsci). Since IOS maps reflect hemodynamic responses driven by sensory evoked activity (coming from thalamus) in cortical neurons, our thinking is that cortical map plasticity and recovery requires the rebuilding of thalamocortical synapse structure and function in tandem with the re-establishment of post-synaptic responses in cortical neurons. Precisely how thalamocortical and intracortical circuits work together to restore function is still an open question and one that deserves more study.

Q8b. Was there a significant correlation between map size/signal and axon turnover?

Response: After reading these reviews and re-examining our submission, we realized that we were not sufficiently clear about how experimental groups were run and depicted in Figure 4a. We sincerely apologize for causing this confusion and have now made it clear in the methods and Figure 4a showing that mice used for imaging axon structure with 2-photon microscopy were separate from those that underwent repeated IOS imaging. Since these experiments were very technically challenging, not to mention physically demanding on the mice (subjected to craniectomy, AAV injections, stroke, daily stimulation, behavioural testing etc), we did not want to subject mice to more than 1 session of anesthesia each week for imaging (eg. 1.5 hours ketamine/xylazine anesthesia for 2-photon imaging and 1.5 hours of isoflurane anesthesia for IOS imaging). Given this situation, we could not report correlations between IOS map size/signal and axonal boutons dynamics.

Q9. Is the partial recovery of the % axon boutons that respond to FL stim significant (Fig. 2c, top of page 4)? This also applies to graphs in Figs. 4f-g, 5b-c, 6b/d and 7a-b: there are always stats on the differences between control and stimulated, but no stats for the effects of stroke

Response: We agree and have now included the stats comparing baseline (ie. before stroke) values to post-stroke values in Chr2 stimulated mice. In Figures 4-7, you'll notice hashtags (#) for all significant comparisons of post-stroke to baseline data and a description of these in each figure legend.

Q10. How do the authors distinguish between the appearance of new boutons vs. the enlargement of pre-existing small boutons (p. 4; Fig. 2e, bouton #4)? Was it always based on mCherry expression and multiple z-slices were checked (instead of deciding based on GCaMP6s expression in a max projection)? What % of newly formed boutons were responsive to FL-stim?

Response: A bouton was considered "new" if its mCherry fluorescence exceeded 2x the median intensity of the shaft, but did not reach this criterion the week before (Note: we checked in multiple z-slices). As for what % of newly formed boutons were responsive to FL stim, we cannot say with certainty because there were so few examples to sample from, but of the few we could find, all appeared to show a GCaMP6s response to FL stimulation.

Q11. Fig. 2e gray bars are sometimes in foreground, sometimes not.

Response: Thank you, we have corrected this.

Q12. "such as beading of cortical dendrites" (p. 5). How do they see dendrites when only axons are labeled? Actually, it would be interesting to see if dendrites are more sensitive to LEDs. (just as they are more sensitive to photodamage and to excitotoxicity). Just curious if they have looked.

Response: Thank you for pointing this out, we meant to say beading of thick axonal projections. We did not examine dendritic structure in our stimulated mice but that may be something worth examining in future studies.

Q13. "...but did NOT induce changes in regional blood flow" (p. 6).

Response: Thank you, good catch.

Q14. The rates of bouton turnover seem a bit higher than those reported by de Paola (2006). The survival fraction of type A1 EPBs reported by de Paola was ~80% after 3-4 weeks. In this study, it's more like 50% in 5 weeks.

Response: Yes the EPB survival rates were higher in the dePaola studies (77-80%) than ours (50-60%) although a major difference is that those studies were focused on PoM or cortical afferents in the normal barrel cortex rather than examining survival rates in stroke damaged tissue.

Q15 Fig. 6a: They should add guides to orient the reader as to where rostral-caudal and medial-lateral directions are.

Response: Done.

Q16. “our estimate of TC bouton responsiveness... is in close agreement with previous work...” (p. 10). This warrants some clarification. How exactly are they comparing chronic calcium imaging of boutons from thalamic neurons using GCaMP6s to acute calcium imaging neuronal somata of L2/3 cortical neurons using OGB? The statement should probably be removed.

Response: We have removed that statement in accord with the reviewer’s suggestion.

Q17. The data on Fig. 7d are underpowered (n=4) to make a reliable conclusion, and the experimental time point for estimating infarct volume does not match the rest of the study (10 days vs. 10 weeks!). Either repeat these infarct volume measurements at 10 weeks post-optogenetics (with more mice) or remove these data.

Response: We agree the sampling in Fig. 7d could have been better to provide a more a reliable conclusion, therefore we have added 2 more mice to each group (n=6 mice / group). We re-analyzed infarct volume in all our mice with an experimenter blind to condition and reached the same conclusion that infarct volumes are not significantly different between the 2 groups (unpaired t-test, p=0.21). This revised data is now shown in the histogram in Figure 7d. The reason we sampled infarct volume at 10 days rather than 10 weeks is that the infarct (glial scar) tends to shrink and is often very difficult to see by 10 weeks post-stroke. By contrast at 10 days, the infarct can be easily defined and measured in more reliable fashion.

Q18. To reduce unnecessary clutter in the main figures, consider moving the following panels to suppl data: 2b (blue panels); 3c; 4d (left diagram: just label the EPB and TB in panel e);

Response: We appreciate the suggestion but we believe Figure 2b and 3c add value to the paper and would like to keep these in the main figures. However, we agree that the cartoon in Fig. 4d is not necessary and have removed it, and labelled EPB and TB in Fig. 4d.

Q19. They should show the different Stroke + control stimulation protocols (with ChR2 but no light, no ChR2 plus light) in suppl. Data.

Response: This is a good suggestion and one raised by reviewer 2 as well. We now include a new analysis and figure that compares thalamocortical bouton turnover, IOS responses and behavioural data in “stroke control stimulation” mice broken down into those mice that expressed ChR2 but did not receive blue light stimulation vs. those did not express ChR2 but received chronic blue light stimulation. As shown in Suppl Figure 6, both groups (ones that received chronic stimulation vs those that didn’t) showed the same pattern of changes after stroke. There were no significant differences between the control groups on any measure (ANOVA Main effect of Group: EPB Turnover, p=0.17; EPB Elimination, p=0.08; EPB Formation, p=0.69; IOS Area, p=0.2; IOS Peak Response, p=0.43; Ladder % Correct, p 0.24; Ladder %

Incorrect, $p=0.28$; FL Asymmetry, $p=0.06$), except for TB dynamics, which may relate to the fact that TB were harder to sample from since they were much fewer in number than EPBs.

Reviewer #2 (Remarks to the Author):

This study teases out the relationship between thalamocortical connections to the forelimb of the somatosensory cortex and an ischemic injury by stroke. The paper argues that optogenetic stimulation of the axons of these connections, which shares characteristics of vibrotactile stimulation at the same frequency, can enhance behavioral recovery. The study argues that this enhancement is due to enhanced turnover of “boutons” on the axons, not due to axonal sprouting or increased dendritic branching. In addition to the increased turnover of thalamocortical boutons, they distinguish between en-passant boutons (EPB) which show an increased turnover in the earlier phases of stroke recovery [2-3 weeks], and terminaux boutons (TB) which show increased turnover in the later phases of stroke recovery [3-5 weeks].

Firstly, the study shows that thalamocortical axons are less responsive to vibrotactile stimuli after stroke, as visualized by GCaMP6s signaling. Secondly, optogenetic stimulation of axons does not increase cerebral blood flow, as studied by laser speckle contrast maps. Thirdly, (as stated above) optogenetic stimulation promoted enhanced thalamocortical axon EPBs and TBs turnover in the 2-3 weeks and 3-5 weeks following stroke respectively, as opposed to controls receiving ChR2 without light, or light without ChR2. Axonal branching was noted not to differ. Lastly, behavioral recovery in horizontal ladder walking and adhesive tape removal following stroke was also enhanced in the optogenetic group. Improvements persisted beyond the optogenetic stimulation and was significantly correlated to EPB formation in weeks 2-3.

The manuscript is novel in that it studied thalamocortical circuitry for somatomotor stroke recovery, rather than cortical neurons that are more commonplace in the literature. Additionally, one of the study's measured output was boutons, rather than larger changes such as axonal branching, allowing smaller changes in structure to be analyzed. The optogenetic stimulation was also applied for much longer, 6 weeks, then other studies which stopped after 4-7 days. One nice control is the results on cerebral blood flow, showing that the improvements observed were not merely due to increased blood flow, which was also shown to be possible with higher frequency of optogenetic stimulation.

Major Criticisms:

Q1. The discussion of EPB vs TB functions argues that EPBs turnover was involved in initiating the behavioral recovery, while TBs may be involved in maintaining this improvement. While the correlation in the results between EPB and enhanced motor recovery supports the first claim, there was no evidence to support the later claim; for example, an experiment stopping the optogenetic stimulation after 2-3 weeks and showing that TB turnover is not enhanced and that the behavioral recovery would not be retained in the long term would elegantly prove this claim. However, if an experiment like this cannot be completed, perhaps a sentence regarding that the function of the TBs remains a mystery.

Response: We agree with the reviewer that a new set of experiments would need to be done in order to clarify the relationship between EPB and TB functions and behavioural recovery and maintenance. However, since these new experiments would not be trivial in scope and are not critical for the understanding of the results in their current form, we have revised our statement about TBs in functional recovery and added a statement (in the discussion) in accord with the reviewer's suggestion that the functional differences between the two bouton types remains a mystery and warrants future investigation.

Q2. Fig 6, the increase in cortical response follow stroke is described as "remapping." in Fig 2, no increase in cortical response was found during recovery period and this discrepancy is not fully discussed. Is the conclusion from IOS results that over recovery the BOLD response becomes stronger independent of activity levels?

Response: We believe that the discrepancy between the two findings can be explained by noting that the data presented in Fig. 2 only refers to mice that underwent a stroke, but had no further intervention to promote recovery. Conversely, the data presented in Fig. 6 show an increase in cortical response area and magnitude in mice that received optogenetic therapy after stroke. We would not conclude that the IOS based response was independent of thalamic activity levels, since of course, the IOS response we measured is based on hemodynamics (which correlates with cortical activity patterns). This sensory evoked cortical hemodynamic response requires activity in both thalamic afferents and intra-cortical neurons to fully elicit the response we see with this imaging technique.

Q3. The statistically equivalent Chr2- and Chr2+ groups (Fig. 5e) suggest that optical illumination itself may be a confounding factor. It would be important to show Chr2- results throughout the study for all measures since stimulation through Chr2 (not indirect effects of warming due to laser light) is the main point of the study.

Response: The reviewer raises a good point about potential effects of optical stimulation, and one that we were very careful about when we started this study. In our initial studies we measured the temperature directly underneath the blue LED throughout the 60 minute stimulation period and did not see any rise in temperature. This is likely due to the low duty cycle of our stimulation approach (5 ms x 5 pulses per second, therefore 2.5% duty cycle). Furthermore to argue against any photo-toxic effects, we imaged thalamocortical axons *in vivo* before and immediately after 60 minutes of intermittent 5Hz light stimulation and found no obvious changes in synaptic structure (Suppl. Fig 5). However in order to provide additional empirical evidence, we now include new analysis and a figure that compare thalamocortical bouton turnover, IOS responses and behavioural data in "stroke control stimulation" mice broken down into those mice that expressed Chr2 but did not receive blue light stimulation vs. those that did not express Chr2 but received chronic blue light stimulation. As shown in Suppl Figure 6, both groups (ones that received chronic stimulation vs those that didn't) showed the same pattern of changes after stroke. Statistically, there wasn't significant differences between groups on any measure (ANOVA Main effect of Group: EPB Turnover, $p=0.17$; EPB Elimination, $p=0.08$; EPB Formation, $p=0.69$; IOS Area, $p=0.2$; IOS Peak Response, $p=0.43$; Ladder % Correct, $p=0.24$; Ladder % Incorrect, $p=0.28$; FL Asymmetry, $p=0.06$), except for TB dynamics, which may relate to the fact that TB were harder to sample from since they were much fewer in number than EPBs. However, as one can see from the TB data in the

Supplementary Figure 6, the light stimulated group was the less dynamic of the control groups, which would argue against a photo-damage effect. Furthermore, we have now plotted all sham stroke data (see Suppl. Fig. 7) showing that there are no major group differences in mice that received chronic optogenetic stimulation vs. those that received no light stimulation. Based on these results and our pilot temperature readings, we are reasonably confident that our results cannot be explained by exposure to the light itself (and possible warming, phototoxicity effects etc).

Minor Criticisms

Q4. In the first section of the results, it is stated that before stroke 25% of GCaMP expressing boutons responded to vibrotactile stimulation and that this number drops significantly after stroke. Please include the percentage that it drops down to in the text (top of page 2) because it would read better.

Response: Done.

Q5. Figure 2c states that boutons recover, but in reality- the last measure (4 weeks post stroke) is not significantly different from the first... although the trend is there.

Response: We have re-phrased the wording in the results and figure legend.

Q6. Figure 2b- the images in blue, upon visual inspection, don't seem to show that the boutons are recovering from 2 weeks to 4 weeks.

Response: We have revised the images presented in Fig. 2b which did not have the correct scale in the original submission. The boutons presented in this example were from the same cortical region but are not the same boutons from week to week (Note: following the exact same boutons expressing GCaMP6s after stroke was just too technically difficult, except in very rare instances such as that shown in Fig. 2e). Therefore judging recovery within the same boutons over time was not possible. However, quantitative assessment of all boutons imaged show there was significant increase in normalized peak calcium responses over time, which is now shown in Suppl. Fig. 4b.

Q7. Figure 5d: the blue shading is not defined in the legend (one assumes that a "stable" bouton is as in figure 5a), and the pre-existing bouton that is removed in the +5 weeks' line, should be colored in red shading (and included in the legend as "eliminated") at the +1-2-week line.

Response: We apologize for any confusion. The blue bouton in Fig. 5d is now defined properly in the figure as a "surviving bouton", meaning that it was present from 1-2 weeks and 5 weeks.

Q8. It is confusing why Figure 5e shows that there are pre-existing boutons at control and stimulated did not have significantly different survival at week 5, while figure 5c shows a significant difference of EPB elimination at week 4.

Response: It is possible that the eliminated boutons in week 4 did not include many pre-existing boutons, and that is why it is not reflected in the week 5 survival rates. Pre-existing boutons are particularly robust, given that they had previously survived the stroke, while newly formed boutons in the intervening weeks were much more likely to be removed at any particular time point.

Q9. Figure 6 is described as "remapping." While this may be true no mapping was actually done, therefore it is unclear if recovery of responses returned in the same topographically organized manner as before. Reword these findings as "functional recovery" of cortex rather than remapping.

Response: Thank you, we have re-worded the results

Reviewer #3 (Remarks to the Author):

This paper shows that trans-cranial optogenetic stimulation of thalamocortical axons (1 hour daily session 5Hz trains; 1 sec on, 4 sec off) is effective in promoting the formation and stabilization of thalamocortical synaptic boutons, the re-emergence of forelimb derived sensory cortical maps, and recovery of sensorimotor abilities. The results are interesting and fairly novel. Previous work has used stimulation of specific brain sites to aid in functional recovery. Some of these studies are cited, but a relevant set of studies that used "contingent" electrical stimulation of the VTA to facilitate functional recovery after FL motor cortex lesion is missing and seems relevant (e.g. 8577374).

Response: We thank the reviewer for reminding us of the study by Castro-Alamancos and Borrell (1995). This was an oversight on our part, since we have cited this important paper in our past studies (Brown et al., 2007, 2009) and it certainly deserves citation in the present one. This reference is specifically referenced in the last paragraph of the discussion when we discuss how to optimize future optogenetic studies.

Q1. The present study employed relatively small infarcts in FLS1. Figure 2 uses GCaMP6s to show that thalamocortical axons are less responsive after stroke. The authors measured calcium responses in axonal boutons before and after the infarct. The imaging portion of this experiment seems straight forward since the brain can be imaged during every session and major features used for alignment. However, what is not at all clear is how they stimulate the forepaw in every session in the same reproducible way. It is possible that the results shown reflect differences in the placement of the stimulation device. What was done to assure that the placement was identical in each session? Note that this is as critical as aligning the imaging part during each session. Ideally, one would use an evoked response that does not change as a result of the manipulation (infarct) and serves to place the paw stimulator. Otherwise, how can we be sure that the stimulus is identical between sessions? Many details are missing about how the stimulation was delivered. For example, was the stimulator only placed in one spot per session? Interpretation of these results and those in figure 6 relies as much on the imaging part as on the stimulus configuration.

Response: The reviewer's comment is well taken and one we were mindful of. The entire forepaw was given vibro-tactile stimulation at 100Hz for 1.5 seconds. This was achieved with an isolated pulse

stimulator (5ms pulse duration with 10ms inter-pulse interval) driving a linear amplifier that was connected to a piezoelectric bending actuator (Piezo Systems, Q220-A4-203YB). The bending actuator has a syringe with a pencil lead in it that we could lightly adhere to the dorsal surface of the forepaw (parallel to the middle digits). Since these mice have dark fur, we could use the slight loss of fur where the pencil lead was attached to re-place the pencil lead in subsequent imaging sessions. For each imaging session, we only used the one placement position since the stimulus was strong enough to vibrate the entire forelimb. This information is now included in the methods section. Based on several years of IOS imaging using this stimulation regime (see Brown et al., 2009, JNsci), we have found that the amplitude and position of cortical responses is reliable from week to week. Furthermore, we have added new IOS imaging data in sham stroke mice (see Fig. 2c and Fig 6b-c) to demonstrate that there were no significant changes in the area and amplitude of cortical responses over several weeks following sham stroke procedure.

Q2. One of the main findings of the study is that bouton TOR is affected by both stroke and optogenetic therapy. It would be useful to show the effects of the same optogenetic therapy in control non-infarcted mice. This seems relevant because it would clarify if the effect observed is just a natural feature of optogenetic stimulation. It would also reveal if normal FLS1 representations (IOS) are affected by this stimulation. In other words, a non-lesioned set of controls (with and without optogenetic stimulation) undergoing the same imaging procedures (IOS and 2P) would be useful. If this is not supplied, the authors should discuss the reasoning for the lack of these relevant groups.

Response: This is an excellent point raised by the other reviewers as well, so we ran a completely new set of sham stroke experiments where we compared TC bouton dynamics, IOS responses and behaviour in sham stroke animals that did not receive optogenetic stimulation (n=5 mice) vs. sham stroke animals that received 6 weeks of optogenetic stimulation, 5 days per week, 1 hour per day, beginning 3 days after sham surgery (n=6 mice). Before getting into the details of the results, the simple answer is that in the absence of stroke (ie. sham stroke) optogenetic stimulation did not have any significant effect on thalamocortical TB and EPB turnover rates, IOS response area or amplitude and behavioural measures of forelimb function (see Suppl. Fig. 7).

In the absence of stroke, there were no significant effects of stimulation group or time in TB turnover rates (Main effect of stimulation: $p = 0.30$; main effect of time: $p = 0.25$; stimulation x time interaction: $p = 0.44$). Similarly, we found no significant effects of stimulation group or time in EPB turnover rates (Main effect of stimulation: $p = 0.95$; main effect of time: $p = 0.14$; stimulation x time interaction: $p = 0.82$). For IOS responses, we found no significant effects of stimulation group or time in the Peak amplitude of forelimb evoked responses (Main effect of stimulation: $p = 0.69$; main effect of time: $p = 0.44$; stimulation x time interaction: $p = 0.50$) or in IOS Response Area (main effect of stimulation: $p = 0.48$; main effect of time: $p = 0.70$; Stimulation x time interaction: $p = 0.08$). Finally, there was no significant effects of optogenetic stimulation group or Time in the Ladder Walking Test (% Correct steps: main effect of stimulation: $p = 0.14$; main effect of time: $p = 0.48$; stimulation x time interaction: $p = 0.89$; % Partial steps: main effect of stimulation: $p = 0.15$; main effect of time: $p = 0.53$; stimulation x time interaction: $p = 0.95$) or Tape Removal Test (forelimb asymmetry: main effect of stimulation: $p = 0.96$; main effect of time: $p = 0.69$; stimulation x time interaction: $p = 0.98$). Based on

these results, we have determined that the beneficial effects of optogenetic stimulation induced by our stimulation protocol are specific to the stroke-affected brain.

Since our sham stroke mice that received chronic optogenetic stimulation were very similar (in all metrics) to those that received control stimulation, we decided to combine these sham stroke data sets together in the main figures so as to reduce visual clutter. However, we have also included all the ungrouped sham stroke data (plots showing TB/EPB turnover, IOS responses and behaviour in sham stimulated vs no stimulation mice) in the Suppl. Fig. 7 for readers to evaluate.

Q3. Related to the previous points, the authors did not seem to attempt to test the effect of optogenetic therapy on either calcium or field potential sensory-evoked responses. Was this attempted? What were the results?

Response: We would love to do this experiment, but technical limitations make this extremely difficult if not impossible in the context of this revision. Unfortunately as other groups have shown (Rickgauer and Tank, 2009, PNAS; Papagiakoumou et al., 2010, Nat Methods; Prakash et al., 2012, Nat Methods), 2-photon scanning at 940nm (the same wavelength that we used to image GCaMP6s) using a 40x objective and comparable or lower laser power density as that used in our study excites and induces photo-currents in ChR2 expressing neurons. This is not surprising since the optimal 2-photon absorption of ChR2 is in the 920-940nm range (~260 GM at 940nm). This unfortunate co-incidence would explain why, to our knowledge, no other group has imaged calcium transients in GCaMP6s expressing neurons that also express ChR2. Recent studies have shown that it could be possible to image GCaMP6 responses in neurons that express a red shifted opsin such as C1V1 (Packet et al., 2015, Nat Methods) or possibly use the blue sensitive ChR2 with a red shifted calcium indicator (RCaMP or JRGEKO). However trying to troubleshoot and implement this novel approach (assuming there was no 2-photon cross talk issues which cannot be guaranteed) would require many months, possibly years to accomplish. Furthermore, we would then have to redo much of the entire study because we would have to prove that all the same optogenetic effects occur with the red shifted opsin. Needless to say, we are definitely interested in this important question and are pushing in that direction, but we think it would be better suited for a future study.

Q4. Figure 4 shows a timeline of experiments. Were the IOS, 2P and behavior done in the same animals? If so, this appears to indicate that animals underwent weekly sessions of isoflurane and ketamine/xylazine anesthesia. This needs to be clarified. If it was the same animals, a description of the timeline of procedures per week should be described.

Response: Animals subjected to behavioural testing also underwent once weekly sessions of either IOS or 2-photon imaging, but not both, in order to reduce the amount of exposure to anesthetic. Additionally, behavioural testing was always conducted the day prior to either IOS or 2-photon imaging so that the maximal amount of time elapsed since the last anesthetic exposure in order to rule out any effects of anesthesia on sensorimotor behaviours. This information is presented in Fig. 4a and more fully described in the associated figure legend.

Q5. The other major finding of the paper is that functional recovery occurs at the behavioral level after optogenetic therapy. It is rather mysterious why a synchronous activation of thalamocortical fibers

at 5 Hz without any relation to behavior would have a beneficial effect on behavior. The effects on boutons seems plausible, but why would such a crude form of stimulation without any relation to ongoing behavior have a beneficial effect? For example, in the functional recovery study noted above, the electrical stimulation had to be contingent to the correct behavior and rewarding. But in this case, the stimulation lacks any relation to ongoing behavior. A different interpretation is that optogenetic stimulation makes the animals more active/alert (or simply “activates” the S1 cortex) and therefore leads to better behavioral outcomes in the tasks. Indeed, the sensory thalamus is known to control the state of cortical activation (20053845), which controls incoming sensory inputs. Such state-related changes would be distinct from those related to circuit rewiring necessary for sensorimotor skills recovery. A discussion of these important issues would be useful.

Response: We would be hesitant to state that light stimulation had no relation to ongoing behaviour. Mice were given stimulation while awake and in an environment where they could ambulate freely and groom etc, so optical stimulation and tactile/proprioceptive sensation would occur in tandem some of the time. As we mention in the discussion, future studies should determine if pairing stimulation with specific phases of forepaw movement would further enhance the recovery of function, similar to what was done in the Alamancos and Borrell (1995) study. With regards to the alternative interpretation that optogenetic stimulation exerts beneficial effects on behaviour by activating the S1 cortex, this is a possibility and one we now consider in the last paragraph of the discussion with reference to the findings of Hirata and Castro-Alamancos, 2010, *JNeurophysiol*. However, since behavioural tests were conducted before the 60 minutes of stimulation, one would not expect an arousal effect to account for improved forepaw use. Furthermore, if this were true then we would have expected to see superior behavioural performance in sham stroke animals given optogenetic stimulation vs. those that did not receive stimulation (see Suppl. Fig 7d-e).

Reviewer #4 (Remarks to the Author):

This is a sophisticated analysis of neural plasticity in the context of recovery after ischemic stroke. Overall, the results are impressive and the experiments are well-designed. Major findings are that stroke reduces number of thalamic axons visible in L1, that there terminal boutons are particularly destabilized, gCaMP imaging in thalamic boutons shows that activity is suppressed in a long term way. The authors then go on to show that repeated optogenetic stimulation of thalamic inputs to L1 can promote functional recovery, using intrinsic signal imaging as well as behavioral measures. The data as presented are beautiful, and the methods are state-of-the-art. However, there are a few gaps in the logic of the paper that would significantly enhance its impact. The effects of optogenetic stimulation on thalamic boutons is well-characterized – and somewhat subtle, making it hard to appreciate the influence altered bouton dynamics will have on thalamocortical transmission – but the effects on gCaMP6 signals is not evaluated. As it stands, the anatomical imaging of boutons is rather anecdotal, and we cannot attribute the change in intrinsic signaling imaging nor the behavioral recovery to a particular process – thalamic transmission or neocortical circuit rearrangement.

Q1a. Control experiments are stroke without stimulation. How dynamic are axons under baseline conditions? If you stimulate axons in the absence of stroke, do you get remodeling? This is important to evaluate the application of the technique to functional enhancement over multiple conditions.

Response: This is an excellent point raised by the other reviewers as well, so we ran a completely new set of sham stroke experiments where we compared TC bouton dynamics, IOS responses and behaviour in sham stroke animals that did not receive optogenetic stimulation (n=5 mice) vs. sham stroke animals that received 6 weeks of optogenetic stimulation, 5 days per week, 1 hour per day, beginning 3 days after sham surgery (n=6 mice). Before getting into the details of the results, the simple answer is that in the absence of stroke (ie. sham stroke) optogenetic stimulation did not have any significant effect on thalamocortical TB and EPB turnover rates, IOS response area or amplitude and behavioural measures of forelimb function (see Suppl. Fig. 7).

More specifically in the absence of stroke, there were no significant effects of stimulation group or time in TB turnover rates (Main effect of stimulation: $p = 0.30$; main effect of time: $p = 0.25$; stimulation x time interaction: $p = 0.44$). Similarly, we found no significant effects of stimulation group or time in EPB turnover rates (Main effect of stimulation: $p = 0.95$; main effect of time: $p = 0.14$; stimulation x time interaction: $p = 0.82$). For IOS responses, we found no significant effects of stimulation group or time in the Peak amplitude of forelimb evoked responses (Main effect of stimulation: $p = 0.69$; main effect of time: $p = 0.44$; stimulation x time interaction: $p = 0.50$) or in IOS Response Area (main effect of stimulation: $p = 0.48$; main effect of time: $p = 0.70$; Stimulation x time interaction: $p = 0.08$). Finally, there was no significant effects of optogenetic stimulation group or Time in the Ladder Walking Test (% Correct steps: main effect of stimulation: $p = 0.14$; main effect of time: $p = 0.48$; stimulation x time interaction: $p = 0.89$; % Partial steps: main effect of stimulation: $p = 0.15$; main effect of time: $p = 0.53$; stimulation x time interaction: $p = 0.95$) or Tape Removal Test (forelimb asymmetry: main effect of stimulation: $p = 0.96$; main effect of time: $p = 0.69$; stimulation x time interaction: $p = 0.98$). Based on these results, we have determined that the beneficial effects of optogenetic stimulation induced by our stimulation protocol are specific to the stroke-affected brain.

Since our sham stroke mice that received chronic optogenetic stimulation were very similar (in all metrics) to those that received control stimulation, we decided to combine these sham stroke data sets together in the main figures so as to reduce visual clutter. However, we have also included all the ungrouped sham stroke data (plots showing TB/EPB turnover, IOS responses and behaviour in sham stimulated vs no stimulation mice) in the Suppl. Fig. 7 for readers to evaluate.

Q1b. Also, because the stroke area can be hard to assess and the window of analysis can vary accordingly, it is important to understand how location of stimulation wrt whether it is closer to normal tissue or in the infarct one related to the degree of plasticity.

Response: The infarct border was assessed 1 week after stroke. At this time, the infarct area is clearly delineated by a lack of small vessels and a white appearance to the tissue, due to the limited blood flow in the area. The 2-photon imaging areas were marked onto a brightfield image of the cortical surface and the distance from the center of each imaging area to the edge of the infarct was measured. The distance was on average $153\mu\text{m}$ with a standard deviation of $88\mu\text{m}$. This information is now included in the methods

section under the heading of imaging axonal structure. As the reviewer points out (and one we have shown in previous studies regarding dendritic spine plasticity; see Brown et al., 2009, JNsci, Reeson et al., 2015, JNsci), the distance from the infarct border is related to the degree of plasticity in a given area. In the present study, we found a significant negative correlation between the distance from the infarct border and the percentage of newly formed boutons in the second and third weeks after stroke, when bouton formation was highest ($r = -0.35$, $p = 0.01$). We should note that there were no differences between the two stroke groups in terms of the mean distance to the infarct border (Stroke + control stim = $153 \pm 84 \mu\text{m}$ vs. Stroke + stimulation = $141 \pm 94 \mu\text{m}$; $p = 0.47$ based on t-test). This data is now reported in the results section. Thus, we suspect that en passant bouton plasticity is heightened in the peri-infarct area, and imaging more than $400 \mu\text{m}$ beyond the infarct border would not yield the same degree of plasticity.

Q2. It is interesting that the investigator makes the tacit assumption that the thalamocortical fibers they are observing must come from the VPM. In fact, thalamic axons in L1 are likely to come from a higher-order thalamic nucleus, such as Pom. This then raises the question of whether VPM axons in L4 and L5b have the same properties as Pom axons.

Response: Although we targeted the AAV injections to the ventral thalamus (eg. VPL), we are cognizant of the fact that virus could not be restricted exclusively to such a small nucleus. As the reviewer notes, the PoM which lies dorsal to the VPL, is also a major source of axons to the FLS1. This statement fits with our cholera toxin labeling experiments in Fig. 1a and connectivity profiles generated by the Allen Brain Institute (<http://connectivity.brain-map.org>). Thus, we agree with the reviewer that the thalamic axons visible in FLS1 are not necessarily from the VPL, but could also originate from the PoM. We have altered the wording in the results section to note this point (see first paragraph in results section) as well as point out retrograde labelling in the PoM in Fig. 1a. The question of whether VPL axons have the same properties as axons from higher-order thalamic nuclei, such as the PoM is very interesting and one that has been studied in the rodent trigeminal system (see Yu, Derdikman, Haidarliu and Ahissar, 2006, PLoS Biology). However, answering this question by imaging calcium responses from each nuclei could easily form the basis of a full length paper and therefore would be better suited for a future study.

Q3. The authors nicely compare the effects of forelimb stimulation to optogenetic activation of thalamic fibers, which have many similarities. However, optogenetic stimulation induces a response with multiple components – probably inhibition, since the synchronized stimulation with light is likely to activate some different intracortical pathways. This difference may be critical in dissecting therapeutic options for thalamic activation in stroke recovery models. How was the light intensity for optogenetic stimulation set – so that it matched the forelimb stim field potential? Or, was it maxed out?

Response: The potential importance of the differing waveforms is an interesting thought and one we had considered. The rationale for setting the light intensity at approximately 10mW/mm^2 was based on the fact that optically evoked cortical response approximated in amplitude the cortical response to forepaw touch. Other reasons were that previous trans-cranial window optogenetic stimulation studies by Karel Svoboda and Josh Trachtenberg's group (see Huber et al., 2008, Nature; Wyatt et al., 2012, Nat NSci) have shown that this blue LED power density reliably activates cortical neurons.

4. In Figure 2, the authors show that thalamic axons are less responsive to forelimb stimulation for weeks after stroke, using signals from gCaMP6. Intrinsic signal imaging in Figure 6 indicates that optogenetic stimulation can induce a change in blood flow. However, the missing piece here is that thalamic axons are becoming more responsive, with stimulation. The anatomical data is all rather descriptive in this regard – it is hard to determine how altered dynamics of EPB or TB will change stimulus-triggered activity in the neocortex. For example, we know that there may not be more boutons, but the axons themselves may be more active. An alternate interpretation is that the neocortex is becoming more responsive, but evoked thalamic activity is staying rather constant. Indeed, the notable plasticity of the neocortex might indicate this is the more likely event. The authors should close the loop on these experiments and show which of these scenarios is true, by doing gCaMP6 imaging before and after prolonged optogenetic stimulation.

Response: We would love to do this experiment, but technical limitations make this extremely difficult if not impossible in the context of this revision. Unfortunately as other groups have shown (Rickgauer and Tank, 2009, PNAS; Papagiakoumou et al., 2010, Nat Methods; Prakash et al., 2012, Nat Methods), 2-photon scanning at 940nm (the same wavelength we used to image GCaMP6s) using a 40x objective and comparable laser power as that used in our study, excites and induces photo-currents in ChR2 expressing neurons. This is not surprising since the optimal 2-photon absorption of ChR2 is in the 920-940nm range (~260 GM at 940nm). This unfortunate co-incidence would explain why, to our knowledge, no other group has imaged calcium transients in GCaMP6s expressing neurons that also express ChR2. Recent studies have shown that it could be possible to image GCaMP6 responses in neurons that express a red shifted opsin such as C1V1 (Packet et al., 2015, Nat Methods) or possibly use the blue sensitive ChR2 with a red shifted calcium indicator (RCaMP or JRGECO). However trying to troubleshoot and implement this novel approach (assuming there was no 2-photon cross talk issues which cannot be guaranteed) would require many months, possibly years to accomplish. Furthermore, we would then have to redo much of the entire study because we would have to prove that all the same optogenetic effects occur with the red shifted opsin. Needless to say, we are definitely interested in this important question and are pushing in that direction, but we think it would be better served in a future paper.

Minor:

1. Scale looks wrong in Figure 3b (too big) – L1 in mouse is typically ~100 μ m.

Response: We re-checked the scale bar and determined it was just slightly too big so we fixed that. We measured layer 1 in this image and it is approximately 80 μ m which fits with what would be expected in post-mortem fixed brain tissue.

2. Figure 3, but especially Figure 3g,h could be supplemental

Response: We would like to keep these in the main figures, since controlling for stimulation induced changes in cerebral blood flow is an important consideration in this study, and one that many scientists whom we have interacted with, have asked for.

Reviewers' comments:

Reviewer #1 (Remarks to the Author):

In their revised manuscript, Tennant and colleagues have painstakingly addressed most of the criticisms I raised. This required a substantial effort. First, they reanalyzed a lot of data, which led to additional interesting results (like finding a decrease in peak amplitude of GCaMP6s DF/F at 1-2 weeks after stroke or a recovery in EPB density with Chr2 after stroke). Second, the authors performed additional experiments, including critical control experiments using a sham stroke. As far as the main criticisms I raised about the original submission of this paper, I am satisfied with their responses to points 1, 2a-d, 3a-c, 5, 6, and 7, but still have lingering concerns regarding my main criticism about the lack of controls (see below). As far as the more minor comments, I am also satisfied with their responses to all, except for Q19 (see below).

- Original point 4a. "Figures 4-7 (and Fig. 2): An important control that is missing is the use of optogenetics in mice with sham stroke (...) to make sure that optogenetic stimulation is not having a non-specific effect on TC axons that is independent of stroke. (...) they must show that in normal mice, Chr2 stimulation of TC boutons has little effect on bouton turnover. Of course this sham stroke control is also important for intrinsic signal imaging (Fig. 6) and behavioral experiments (Fig. 7)."

I give a lot of credit to the authors for performing the sham stroke control experiments, but I have two remaining concerns. The first one is less serious and has to do with the fact that they chose to do 'Rose Bengal without green light' as a sham stroke, whereas the better control would have been, 'green light without Rose Bengal' (since prolonged green light exposure for 15 min could have had some effects on bouton dynamics). However, because in my review I had not asked for specifically for the latter (I only pointed out it would be the better option), I am satisfied with their efforts to address this concern. The second and more serious concern is that they decided to pool all the sham stroke data (with and without Chr2 stimulation). This doesn't make sense for Figs. 4-7. The issue that reviewers raised was to rule out the possibility that optogenetic stimulation is not having a non-specific effect on TC axons (or IOS maps, or behavior) that is independent of stroke. Pooling the data prevents the reader from making this comparison. Rather than showing the separate data in Suppl. Fig. 7, they should show all 4 groups in the main figures. The reader will be especially curious about comparisons between Sham Stroke with Chr2 Stim and Sham Stroke with Control Stim and between Sham Stroke and Chr2 at every point vs. its own baseline (the # stats). My concern, unfortunately, is that the data in Suppl. Fig. 7a seem to show that there is a transient effect of Chr2 in sham stroke mice on TB TOR at 2 weeks (due to an apparent increase in TB formation). A similar effect of Chr2 stim on Sham Stroke mice can be seen on the IOS map area at day +2 (Suppl., Fig. 7c). Only statistics can determine whether these effects are significant. For the other panels in Suppl. Fig. 7, it looks like Chr2 Stim had little effect on Sham Stroke mice for EPB dynamics and for behavior, so I think the authors conclusions will stand for those, but they should show detailed stats for all comparisons in a Suppl. Table.

- Original point 4b. "Moreover, they should show that optogenetics restores to normal the fraction of axon boutons that respond to FL stimulation in peri-infarct cortex?"

I am not sure I agree that 2P imaging parameters to record GCaMP6s signals in boutons would be sufficient to trigger ChR2. I also don't buy the argument that it has not been done yet, because I can't think of why people would have looked (other than to do this experiment). It would have been simple enough to check: do you trigger GCaMP6s signals when you scan with 2P. I do agree that pursuing these experiments with a red-shifted GECI is beyond the scope of this manuscript. Still, without this experiment it's impossible to argue how exactly ChR2 stimulation might lead to changes in behavioral recovery and axon bouton turnover after stroke. This caveat should be stated in the Discussion.

- Original point 19. "They should show the different Stroke + control stimulation protocols (with ChR2 but no light, no ChR2 plus light) in suppl. Data."

Again, I credit the authors for showing these data in Suppl. Fig. 6, but this raises some concerns because of the variability in the data, especially the TB turnover, between the two stroke groups (ChR2 without light vs. no ChR2 + light); the two sometimes differ by > 50%. The same can be said about the FL asymmetry data. Indeed, they found significant differences between these groups on TB dynamics (which cannot easily be explained), and other p values were very close to the arbitrary 0.05 significance value. If the TB dynamic measurements are unreliable, as the authors suggest in their rebuttal, perhaps it is best to remove all data related to TB from the paper. At this point, I am not confident that their conclusions for Fig. 4f-h will stand the test of time.

In summary, I think the revised manuscript is very much improved with the extensive revisions and additional data provided. But I do worry that the appropriate statistical comparisons have not been made in the main figures (e.g., missing Sham Stroke with ChR2 Stim vs. Sham Stroke without Stim). If the authors show all 4 group comparisons in the figures and remove the data on TB dynamics (or move it to Suppl. Data given the caveats about how precise/reliable those measurements are), I believe the bulk of their conclusions will stand and their paper will be a fantastic contribution.

Reviewer #2 (Remarks to the Author):

This revised manuscript addresses the original concerns of this reviewer. This is a substantial piece of work and one that will help inform the field.

One minor note is that there is a recognized beading in dendrites exposed to ischemia that may be confusing to readers associated with studies in neuronal cell death, as regard to EB boutons. It may be worth noting that the observation points in this study are sufficiently late after the ischemic stimulus that these swellings are indeed likely to be EB presynaptic structures and not beaded dendrites.

Reviewer #3 (Remarks to the Author):

The authors have clarified most of the points I raised. I have no additional comments.

Reviewer #4 (Remarks to the Author):

I am satisfied with the revisions. I recommend acceptance.

Dear editors and reviewers:

We would like to thank you for the opportunity to revise our manuscript titled “**Optogenetic rewiring of thalamocortical circuits to restore function in the stroke injured brain**” for publication in *Nature Communications*. We very much appreciate the reviewers’ efforts in this process and their supportive comments on our work. We have now addressed the remaining queries raised by reviewer 1, specifically moving terminaux bouton (TB) data to the Supplementary section and presenting both sham stroke groups (i.e. those with Chr2 stimulation and those without) in the main figures with appropriate statistical analyses.

Reviewer #1:

In their revised manuscript, Tennant and colleagues have painstakingly addressed most of the criticisms I raised. This required a substantial effort. First, they reanalyzed a lot of data, which led to additional interesting results (like finding a decrease in peak amplitude of GCaMP6s DF/F at 1-2 weeks after stroke or a recovery in EPB density with Chr2 after stroke). Second, the authors performed additional experiments, including critical control experiments using a sham stroke. As far as the main criticisms I raised about the original submission of this paper, I am satisfied with their responses to points 1, 2a-d, 3a-c, 5, 6, and 7, but still have lingering concerns regarding my main criticism about the lack of controls (see below). As far as the more minor comments, I am also satisfied with their responses to all, except for Q19 (see below).

Query #1: I am not sure I agree that 2P imaging parameters to record GCaMP6s signals in boutons would be sufficient to trigger Chr2. I also don’t buy the argument that it has not been done yet, because I can’t think of why people would have looked (other than to do this experiment). It would have been simple enough to check: do you trigger GCaMP6s signals when you scan with 2P. I do agree that pursuing these experiments with a red-shifted GECI is beyond the scope of this manuscript. Still, without this experiment it’s impossible to argue how exactly Chr2 stimulation might lead to changes in behavioral recovery and axon bouton turnover after stroke. This caveat should be stated in the Discussion.

Response to Query #1: As the reviewer notes, one could check to see if GCaMPs fluorescence levels were abnormally high in Chr2 expressing mice, and in fact, we did find significantly higher median GCaMP6s fluorescence in pilot experiments, despite keeping our imaging parameters within the normal range. This observation, coupled with the elegant published work from David Tank (Rickgauer and Tank, 2009, PNAS) as well as Karl Deisseroth’s lab (Prakash et al., 2012, Nature Methods) showing that 920-940nm 2-photon scanning leads to photocurrents in Chr2 expressing neurons, led to us to the conclusion that this experiment should be left for a future study with optogenetic probes that do not have significant 2-photon cross talk (i.e. pursued with red-shifted GECI or Chr2). Therefore, as the reviewer suggests, we now state in the discussion (page 12) that we do not know exactly how Chr2 stimulation is affecting axonal excitability and that future studies should address this question.

Query #2:

I give a lot of credit to the authors for performing the sham stroke control experiments, but I have two remaining concerns. The first one is less serious and has to do with the fact that they chose to do 'Rose Bengal without green light' as a sham stroke, whereas the better control would have been, 'green light without Rose Bengal' (since prolonged green light exposure for 15 min could have had some effects on bouton dynamics). However, because in my review I had not asked for specifically for the latter (I only pointed out it would be the better option), I am satisfied with their efforts to address this concern. The second and more serious concern is that they decided to pool all the sham stroke data (with and without Chr2 stimulation). This doesn't make sense for Figs. 4-7. The issue that reviewers raised was to rule out the possibility that optogenetic stimulation is not having a non-specific effect on TC axons (or IOS maps, or behavior) that is independent of stroke. Pooling the data prevents the reader from making this comparison. Rather than showing the separate data in Suppl. Fig. 7, they should show all 4 groups in the main figures. The reader will be especially curious about comparisons between Sham Stroke with Chr2 Stim and Sham Stroke with Control Stim and between Sham Stroke and CHR2 at every point vs. its own baseline (the # stats). My concern, unfortunately, is that the data in Suppl. Fig. 7a seem to show that there is a transient effect of Chr2 in sham stroke mice on TB TOR at 2 weeks (due to an apparent increase in TB formation). A similar effect of Chr2 stim on Sham Stroke mice can be seen on the IOS map area at day +2 (Suppl., Fig. 7c). Only statistics can determine whether these effects are significant. For the other panels in Suppl. Fig. 7, it looks like Chr2 Stim had little effect on Sham Stroke mice for EPB dynamics and for behavior, so I think the authors conclusions will stand for those, but they should show detailed stats for all comparisons in a Suppl. Table.

Query #3:

Again, I credit the authors for showing these data in Suppl. Fig. 6, but this raises some concerns because of the variability in the data, especially the TB turnover, between the two stroke groups (Chr2 without light vs. no Chr2 + light); the two sometimes differ by > 50%. The same can be said about the FL asymmetry data. Indeed, they found significant differences between these groups on TB dynamics (which cannot easily be explained), and other p values were very close to the arbitrary 0.05 significance value. If the TB dynamic measurements are unreliable, as the authors suggest in their rebuttal, perhaps it is best to remove all data related to TB from the paper. At this point, I am not confident that their conclusions for Fig. 4f-h will stand the test of time.

In summary, I think the revised manuscript is very much improved with the extensive revisions and additional data provided. But I do worry that the appropriate statistical comparisons have not been made in the main figures (e.g., missing Sham Stroke with Chr2 Stim vs. Sham Stroke without Stim). If the authors show all 4 group comparisons in the figures and remove the data on TB dynamics (or move it to Suppl. Data given the caveats about how precise/reliable those measurements are), I believe the bulk of their conclusions will stand and their paper will be a fantastic contribution.

Response to Query #2 and 3: The reviewer has a good point (Query #2) in that we should show the sham stroke control groups (with and without optogenetic stimulation) individually in Main Figures 5-7 and Supplementary Fig. 7. Therefore, we now present this data in the main figures of the revised manuscript (see solid black and dashed grey lines in Figs. 5-7) with statistical comparisons to BL data (# stats) when appropriate. Our analysis indicates there was no significant main effects of ChR2 stimulation in the sham stroke control groups on any dependent measure. This is reported in the results section on page 7.

Regarding the reviewer's comment (under Query #2): "The reader will be especially curious about comparisons between Sham Stroke with ChR2 Stim and Sham Stroke with Control Stim and between Sham Stroke and ChR2 at every point vs. its own baseline (the # stats). My concern, is that the data in Suppl. Fig. 7a seem to show that there is a transient effect of ChR2 in sham stroke mice on TB TOR at 2 weeks (due to an apparent increase in TB formation). A similar effect of ChR2 stim on Sham Stroke mice can be seen on the IOS map area at day +2 (Suppl., Fig. 7c)." As requested by the reviewer we now present statistical comparisons for each sham stroke control group to their respective baseline values (see color coded #) when justified by a significant ANOVA. Regarding the transient increase in TB TOR at 3 weeks (due to an increase in TB elimination), this was found in the sham stroke controls that did not receive ChR2 stimulation (black lines in Supp Fig. 7). Since there was a significant Group x Time interaction, we show the t-test comparing Week 3 vs baseline in Supp Fig. 7c-d. As for the IOS map area at day 2, we did not find any significant effect of Group, Time or Group x Time Interaction on FL map area in sham stroke controls. Given this, we could not justify comparing the day 2 timepoint to baseline (Note: ChR2 stimulation began on day 3 in this group so it would not be a factor). However, even if we went ahead and made this statistical comparison, it was not significant (based on t-test).

As requested under query #3, we have now have moved the TB dynamics data (Fig 4d-h in previous revision) to the Supplementary data section (see new Supp. Fig. 7). We have also added a sentence to the results (page 8) mentioning the variability issue and caveats associated with TB data analysis. The variability in the asymmetry score (Fig. 7b) is expected based on our own experience with this behavioural test, and similar to other papers that report asymmetry scores after stroke (e.g. MacLellan et al., 2006, JCBFM; Martinez et al., 2011, PLoS One).

Reviewer #2 (Remarks to the Author):

This revised manuscript addresses the original concerns of this reviewer. This is a substantial piece of work and one that will help inform the field.

Q1. One minor note is that there is a recognized beading in dendrites exposed to ischemia that may be confusing to readers associated with studies in neuronal cell death, as regard to EB boutons. It may be worth noting that the observation points in this study are sufficiently late after the ischemic stimulus that these swellings are indeed likely to be EB presynaptic structures and not beaded dendrites.

Response: This reviewer is correct in noting that dendritic structures can appear beaded in the acute phase after stroke. However, because we injected AAV into the thalamus and imaged these axons in the cortex, we did not encounter labelled cortical dendrites in the FLS1 cortex. We have added a sentence in the results (page 7) stating that: “We should note that since AAV infected neurons in the thalamus, we did not encounter fluorescently labelled cortical dendrites when imaging. This mitigated any confusion in distinguishing beaded pre-synaptic structures with ischemia related beading of post-synaptic dendrites (that can occur in the first 72 hours after stroke).”

Reviewer #3 (Remarks to the Author):

The authors have clarified most of the points I raised. I have no additional comments.

Reviewer #4 (Remarks to the Author):

I am satisfied with the revisions. I recommend acceptance.

REVIEWERS' COMMENTS:

Reviewer #1 (Remarks to the Author):

I am satisfied with the authors' response to my concerns. I have no further comments. I believe the revisions have strengthened the quality of this paper.